# Instability in Generative Adversarial Imitation Learning with Deterministic Policy

## Abstract

Deterministic policies are widely applied in generative adversarial imitation learning (GAIL). When adopting these policies, some GAIL variants modify the reward function to avoid training instability. However, the mechanism behind this instability is still largely unknown. In this paper, we capture the instability through the underlying exploding gradients theoretically in the updating process. Our novelties lie in: 1) We establish and prove a probabilistic lower bound for exploding gradients, which can describe the instability universally, while the stochastic policy will never suffer from such pathology subsequently, by employing the multivariate Gaussian policy with small covariance to approximate deterministic policy. 2) We also prove that the modified reward function of adversarial inverse reinforcement learning (AIRL) can relieve exploding gradients. Experiments support our analysis.

## 1 Introduction

Imitation learning (IL) trains a policy directly from expert demonstrations without reward signals (Ng et al., 2000; Syed & Schapire, 2007; Ho & Ermon, 2016). It has been broadly studied under the twin umbrellas of behavioral cloning (BC) (Pomerleau, 1991) and inverse reinforcement learning (IRL) (Ziebart et al., 2008). Generative adversarial imitation learning (GAIL) (Ho & Ermon, 2016), established by the trust region policy optimization (TRPO) (Schulman et al., 2015) policy training, plugs the inspiration of generative adversarial networks (GANs) (Goodfellow et al., 2014) into the maximum entropy IRL. The discriminator in GAIL aims to distinguish whether a state-action pair comes from the expert demonstration or is generated by the agent. Meanwhile, the learned policy generates interaction data for confusing the discriminator. GAIL is promising for many real-world scenarios where designing reward functions to learn the optimal control policies requires significant effort. It has made remarkable achievements in physical-world tasks, e.g., robot manipulation (Jabri, 2021), mobile robot navigating (Tai et al., 2018), commodities search (Shi et al., 2019), endovascular catheterization (Chi et al., 2020), etc.

The learned policy in GAIL can be effectively accomplished by reinforcement learning (RL) methods (Sutton & Barto, 2018; Puterman, 2014), which are divided into stochastic policy algorithms and deterministic policy algorithms, incorporating the two classes of algorithms into GAIL denoted as ST-GAIL and DE-GAIL respectively in our paper. For ST-GAIL, one can refer to the proximal policy optimization (PPO)-GAIL (Chen et al., 2020), the natural policy gradient (NPG)-GAIL (Guan et al., 2021) and the two-stage stochastic gradient (TSSG) (Zhou et al., 2022). These algorithms have shown that GAIL can ensure global convergence in high-dimensional environments against traditional IRL methods (Ng et al., 2000; Ziebart et al., 2008; Boularias et al., 2011). Unfortunately, ST-GAIL methods have low sample efficiency and cost a long time to train the learned policy well (Zuo et al., 2020).

In comparison, some related works (Kostrikov et al., 2019; Zuo et al., 2020) imply that deterministic policies are capable of enhancing sample efficiency when training GAIL variants. Kostrikov et al. (2019) proposed the discriminator-actor-critic (DAC) algorithm, which defines the reward function with the GAIL discriminator for the policy trained by the twin delayed deep deterministic policy gradient (TD3) (Fujimoto et al., 2018). It reduces policy-environment interaction sample complexity by an average factor of 10. Deterministic generative adversarial imitation learning (DGAIL) (Zuo

et al., 2020) utilizes the modified deep deterministic policy gradient (DDPG) (Lillicrap et al., 2015) to update the policy, and achieves a faster learning speed than ST-GAIL.

These DE-GAIL methods not only effectively improve sample efficiency, but also mitigate instability caused by the reward function $-\log(1 - D(s, a))$, referred to as the *positive logarithmic reward function* (PLR). This classical logarithmic reward function is one of the primary shapes and is frequently used in GAIL. For PLR, Kostrikov et al. (2019) discussed the reward bias in a simple toy demo; Zuo et al. (2020) exhibited its instability through contrast experiments, and improved the stability of DE-GAIL by introducing the idea of learning from demonstrations (LfD) into the generator.

However, the instability caused by PLR is largely unknown. We investigate the performance of DDPG-GAIL and TD3-GAIL, i.e., replacing TRPO to RL algorithms DDPG and TD3 with unchanged PLR, with $10^6$ expert demonstrations displayed in Fig. 1. It emerges extreme instability under 11 random seeds. The learning effects under some seeds are capable of reaching expert levels (valid), while others hardly learn anything (invalid). At some point, the two corner cases cannot be well captured by the bias analysis introduced through the toy demo in Kostrikov et al. (2019).

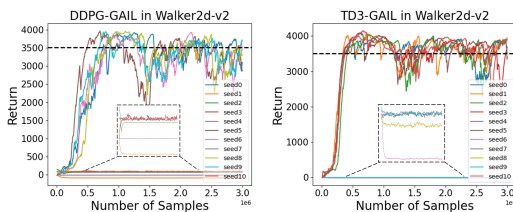

Figure 1: Learning curves of DDPG-GAIL and TD3-GAIL with different random seeds in Walker2d-v2.

In this paper, we incorporate a different deterministic policy algorithm, i.e., the softmax deep double deterministic policy gradients (SD3) (Pan et al., 2020) into GAIL with PLR. SD3-GAIL exhibits the same instability as the experiment results in DDPG-GAIL and TD3-GAIL. This implies that instability is not a special case in DE-GAIL. In addition, the instability of DE-GAIL is mainly down to the invalid case that appeared in the experiment. Then, we prove that there exist exploding phenomena in the absolute gradients of the policy loss, describing the invalidity theoretically and universally. Further, we conclude that the discriminator will possibly degenerate to 0 or 1. Meanwhile, we give the probabilistic lower bound for exploding gradients, with respect to the mismatching between the learned policy and the expert demonstration, i.e., the significant state-action distribution discrepancy between the learned state-action pair and that of the expert. Finally, we disclose that the outliers interval under the modified reward function in adversarial inverse reinforcement learning (AIRL) (Fu et al., 2018) is smaller than that under PLR in GAIL. Such a modified reward function shows its superiority of stability. Our contributions can be summarized as follows:

- Establish and prove the probability of exploding gradients theorem to theoretically describe the instability in DE-GAIL. In contrast, our analysis strikes that stochastic policy can avoid exploding gradients.
- Compared with the consistent trend in ST-GAIL, we point out the instability is caused by deterministic policies, rather than GANs.
- The reward function in AIRL is shown to reduce the probability of exploding gradients.

## 2 RELATED WORK

In large and high-dimensional environments, Ho & Ermon (2016) proposed GAIL, which is processed on TRPO (Schulman et al., 2015). It gains significant performance in imitating complex expert policies (Ghasemipour et al., 2019; Xu et al., 2020; Ke et al., 2020; Chen et al., 2021).

To accelerate the GAIL learning process, a natural idea is to use deterministic policy gradients. Sample-efficient adversarial mimic (SAM) (Blondé & Kalousis, 2019) method integrates DDPG (Lillicrap et al., 2015) into GAIL, and adds a penalty on the gradient of the discriminator meanwhile. Zuo et al. (2020) proposed deterministic generative adversarial imitation learning (DGAIL), which combines the modified DDPG with LfD to train the generator under the guidance of the discriminator. The reward function in DGAIL is set as $D(s, a)$. TD3 (Fujimoto et al., 2018) and off-policy training of the discriminator are performed in DAC (Kostrikov et al., 2019) to reduce policy-environment interaction sample complexity by an average factor of 10. The revised reward function in DAC is $\log(D(s, a)) - \log(1 - D(s, a))$.

Notably, these works achieve gratifying results benefiting from modifications to reward functions. When implementing GAIL with PLR directly, Kostrikov et al. (2019) pointed out the reward bias via a special toy demo; DGAIL exhibited its instability merely from the perspective of experimental results. Differently, we illustrate this phenomenon through a universal theory.

## 3 PRELIMINARY

In this section, we introduce the definition of a Markov decision process, reproducing kernel Hilbert space and generative adversarial imitation learning setup.

### 3.1 MARKOV DECISION PROCESS

A discounted Markov decision process (MDP) is characterized by a 5-tuple $(\mathcal{S}, \mathcal{A}, r, p_M, \gamma)$ in the standard RL setting. $\mathcal{S}$ and $\mathcal{A}$ denote the finite state space and action space, respectively. $r(s, a) : \mathcal{S} \times \mathcal{A} \to \mathbb{R}$ is the reward function for performing action $a \in \mathcal{A}$ in state $s \in \mathcal{S}$. $p_M(s'|s, a) : \mathcal{S} \times \mathcal{A} \times \mathcal{S} \to [0, 1]$ denotes the transition distribution and $\gamma$ is the discount factor. A policy $\pi(a|s)$ specifies an action distribution conditioned on state $s$. The objective of RL is to maximize the expected reward-to-go $\eta(\pi) = \mathbb{E}_\pi \left[ \sum_{t=0}^{\infty} \gamma^t r(s_t, a_t) |s_0, a_0 \right]$.

Induced by a policy $\pi$, we define *the discounted stationary state distribution* as

$$d^\pi(s) = (1 - \gamma) \sum_{t=0}^{\infty} \gamma^t \Pr(s_t = s; \pi).$$

Here $\Pr(s_t = s; \pi)$ denotes the probability of reaching state $s$ at time $t$, which is given by

$$\int \prod_{u=0}^{t-1} p_M\left(s_{u+1} \mid s_u, a_u\right) \pi\left(a_u \mid s_u\right) \Pr\left(s_0\right) d\boldsymbol{s} d\boldsymbol{a},$$

where $d\boldsymbol{s} = ds_0 \dots ds_{t-1}$ and $d\boldsymbol{a} = da_0 \dots da_{t-1}$ imply integration over the previous states and actions. Similarly, *the discounted stationary state-action distribution* is defined as

$$\rho^\pi(s, a) = (1 - \gamma) \sum_{t=0}^{\infty} \gamma^t \Pr(s_t = s, a_t = a; \pi),$$

which measures the overall "frequency" of visiting a state-action pair under the policy $\pi$. The relationship between $\rho^\pi(s, a)$ and $d^\pi(s)$ can be described as

$$\rho^\pi(s, a) = \pi(a|s) d^\pi(s). \tag{1}$$

### 3.2 REPRODUCING KERNEL HILBERT SPACE

Given a set $\mathcal{S}$, for any $c \in \mathbb{R}^p$ and $x \in \mathcal{S}$, if the linear functional mapping $h \in \mathcal{H}$ to $(c, h(x))$ is continuous, then a vector-valued reproducing kernel Hilbert space (RKHS) $\mathcal{H}$ is a Hilbert space of functions $h : \mathcal{S} \to \mathbb{R}^p$ (Micchelli & Pontil, 2005). For all $y \in \mathcal{S}$, $\kappa_x(y)$ is a symmetric function that is a positive definite matrix, thereby $(\kappa_x c)(y) = \kappa(x, y)c \in \mathcal{H}$. It has the reproducing property $\langle h, \kappa_x c \rangle_{\mathcal{H}} = h(x)^\top c$. Here $(\cdot, \cdot)$ and $\langle \cdot, \cdot \rangle_{\mathcal{H}}$ denote the inner product in $\mathbb{R}^p$ and in $\mathcal{H}$, respectively. We denote $\mathcal{H} = \mathcal{H}_\kappa$.

### 3.3 GENERATIVE ADVERSARIAL IMITATION LEARNING

GAIL combines IRL with GANs, treating RL methods as the generator. GAIL takes the advantage of the discriminator $D(s, a)$ to calculate the difference between the distribution of the state-action pair induced by the learned policy $\pi$ and the expert policy $\pi_E$, thereby providing the reward for the agent. Moreover, the policy and discriminator can be approximated by RKHS (Ormoneit & Sen, 2002). The optimization problem in GAIL is

$$\min_\pi \max_{D \in (0,1)^{\mathcal{S} \times \mathcal{A}}} \mathbb{E}_{(s,a) \sim \rho^{\pi_E}} [\log(D(s, a))] + \mathbb{E}_{(s,a) \sim \rho^\pi} [\log(1 - D(s, a))], \tag{2}$$

where the policy $\pi$ mimics the expert policy via the reward function $r(s,a) = -\log(1 - D(s,a))$. When the discriminator reaches its optimal

$$D^*(s,a) = \rho^{\pi_E}(s,a)/(\rho^{\pi_E}(s,a) + \rho^{\pi}(s,a)), \tag{3}$$

the optimization objective of the learned policy is formalized as minimizing the state-action distribution discrepancy between the imitated policy and the expert policy with the Jensen-Shannon (JS) divergence:

$$\min_{\pi} D_{\text{JS}}(\rho^{\pi}(s,a), \rho^{\pi_E}(s,a)) := \frac{1}{2} D_{\text{KL}}\left(\rho^{\pi}, \frac{\rho^{\pi} + \rho^{\pi_E}}{2}\right) + \frac{1}{2} D_{\text{KL}}\left(\rho^{\pi_E}, \frac{\rho^{\pi} + \rho^{\pi_E}}{2}\right). \tag{4}$$

## 4 THE PRINCIPLE OF INSTABILITY IN GAIL WITH DETERMINISTIC POLICIES

Inspired by the instability of DE-GAIL in MuJoCo environments (Subsect. 4.1), we theoretically impute the invalidity to exploding phenomenon in the absolute gradients of the policy loss in Subsect. 4.2. Additionally, the probabilistic lower bound for exploding gradient is provided. In contrast, ST-GAIL will not suffer from such pathology. Finally, we present that the reward function in AIRL relieves the exploding gradients in Subsect. 4.3. Fig. 2 shows the architecture for our analysis.

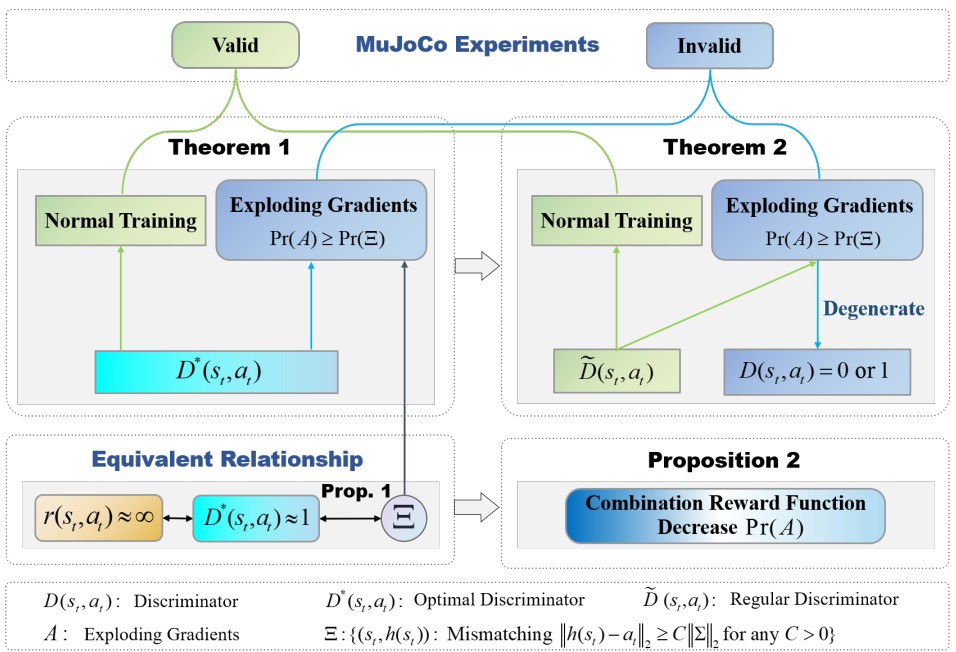

Figure 2: An illustration of our proposed theorem of exploding gradients in DE-GAIL. Due to exploding gradients, the invalid case in experiments corresponds to diverging from the optimal discriminator or degenerating to 0 or 1.

### 4.1 ILLNESS REWARD EMPIRICAL EVALUATIONS IN MUJOCO ENVIRONMENTS

We replicate the experimental setup of Zhou et al. (2022). Expert trajectories are created by the SAC agent in Hopper-v2, HalfCheetah-v2 and Walker2d-v2 respectively. The size of expert demonstration data is $10^6$ obtained with a 0.01 standard deviation. The mean return of the demonstration data in each environment is 3433, 9890 and 3509 respectively.

When training GAIL, we use two-layer networks to approximate the kernel function (see Arora et al. (2019)). The reward function is set as $r(s,a) = -\log(1 - D(s,a))$ (PLR). First, a well-performed TSSG is revisited as the baseline for subsequent comparison. Then we explore DDPG and TD3, which also cause GAIL to fail as Kostrikov et al. (2019). A brief description of the

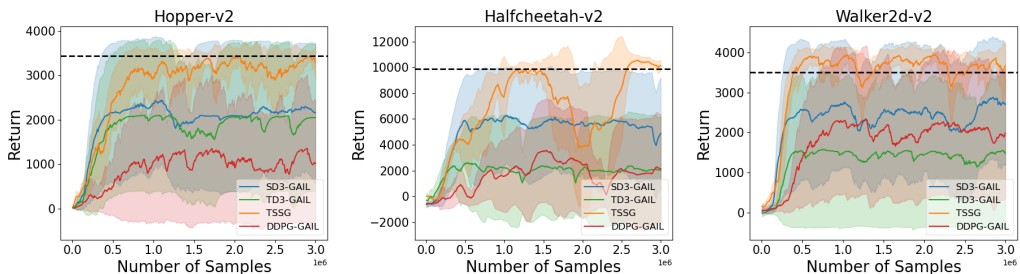

Figure 3: Comparison of SD3-GAIL, TD3-GAIL, DDPG-GAIL and TSSG in three different environments. Solid lines and dashed lines correspond to the average performance of the four algorithms and expert demonstrations, respectively.

training procedure is laid out in Appendix A.1 (Alg. 1). The best evaluation results of DDPG-GAIL and TD3-GAIL are shown in Fig. 3. Compared to TSSG, we can observe that DDPG-GAIL and TD3-GAIL not only obtain a lower mean return but also suffer from obviously higher variance. We suspect this is owing to the inaccurate Q-value estimations of DDPG and TD3 (Pan et al., 2020).

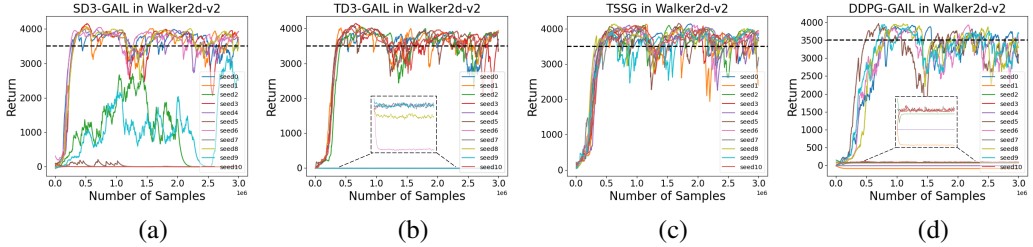

Figure 4: The performances of the four algorithms under different 11 seeds in Walker2d-v2.

We next experimentally examine whether better value estimation will improve GAIL. SD3 (Pan et al., 2020), a deterministic policy algorithm that enables a smaller absolute bias of true values and value estimates than TD3, is conducted into the framework of GAIL, as shown in Alg. 1. Its best performances are presented in Fig. 3. We conclude the results in two aspects:

- SD3-GAIL improves the average return compared to DDPG-GAIL and TD3-GAIL, and even achieves a higher upper limit than TSSG. This phenomenon is possibly attributed to the accuracy of Q-value estimation in SD3.

- Although the enhancement of average return in SD3-GAIL compared to DDPG-GAIL and TD3-GAIL, the variance of SD3-GAIL remains high. In order to thoroughly observe the phenomenon of high variance of SD3-GAIL, DDPG-GAIL and TD3-GAIL, we specify the performance of the four algorithms under 11 random seeds in Walker2d-v2 as exhibited in Fig. 4. The three deterministic policy algorithms reveal extreme instability under 11 random seeds. Specifically,

  - The performances under some seeds are capable of reaching expert levels, such as SD3-GAIL, TD3-GAIL and DDPG-GAIL with seed 0.
  - Some seeds almost learn nothing, such as SD3-GAIL, TD3-GAIL and DDPG-GAIL with seed 7.
  - SD3-GAIL with seed 2 and seed 9 learn only part of the expert policy.

  While TSSG maintains a similar trend under different random seeds, the instability is not caused by GANs itself depicted by Arjovsky & Bottou (2017).

These empirical evaluations manifest that DE-GAIL algorithms suffer from pathological training dynamics. We attempt to explain this phenomenon by employing the statement of reward bias introduced by Kostrikov et al. (2019). In their specific toy demo, repeated trajectories may be trained with the stochastic learned policy, which brings about a higher return than expert under PLR

$r(s,a) = -\log(1 - D(s,a))$, the strictly positive reward function. This is known as the reward bias. Conversely, if the learned policy is deterministic, the trajectories will be trained with no repetition. Furthermore, the expert return will not be exceeded by the return of the deterministic learned policy technically, in other words, the reward bias vanishes. This can not interpret DE-GAIL as inferior to ST-GAIL which appeared in our experiment. An illustrative example of the invalid case is shown in Appendix A.2. In the sequel, we will explain such instability from the perspective of invalidity in DE-GAIL.

## 4.2 EXPLODING GRADIENTS IN DE-GAIL

Inspired by Lever & Stafford (2015) and Paternain et al. (2020) who employ multivariate Gaussian policy to approximate deterministic policy, we define the learned policy $\pi_h$ as

$$\pi_h(a|s) = \frac{1}{\sqrt{\det(2\pi\boldsymbol{\Sigma})}} \exp \frac{-(a - h(s))^\top \boldsymbol{\Sigma}^{-1}(a - h(s))}{2},$$

parameterized by deterministic functions $h \in \mathcal{H}, h : \mathcal{S} \to \mathcal{A}$ and covariance matrix $\boldsymbol{\Sigma}$. The function $h(\cdot)$ is an element of an RKHS $\mathcal{H}_\kappa$, $h(\cdot) = \sum_i \kappa(s_i, \cdot)a_i \in \mathcal{H}_\kappa$, where $s_i \in \mathcal{S}$ and $a_i \in \mathcal{A}$. Note that $\pi_h(a|s)$ can be regarded as an approximation to the Dirac's impulse via covariance matrix approaching zero, i.e.,

$$\lim_{\boldsymbol{\Sigma} \to \boldsymbol{0}} \pi_h(a|s) = \delta(a - h(s)). \tag{5}$$

Eq. (5) means that when the covariance $\boldsymbol{\Sigma} \to \boldsymbol{0}$, the stochastic policy $\pi_h(a|s)$ approaches the deterministic policy $h(s)$. Replacing $\pi$ with $\pi_h$ in Eq. (2), Eq. (3) and Eq. (4) respectively, the optimization problem of GAIL under $\pi_h$ is

$$\min_{\pi_h} \max_D \ \mathbb{E}_{(s,a) \sim \rho^{\pi_E}}[\log(D(s,a))] + \mathbb{E}_{(s,a) \sim \rho^{\pi_h}}[\log(1 - D(s,a))],$$

the optimum discriminator is

$$D^*(s,a) = \rho^{\pi_E}(s,a)/(\rho^{\pi_E}(s,a) + \rho^{\pi_h}(s,a)), \tag{6}$$

and the policy optimization objective is

$$\min_{\pi_h} D_{\mathrm{JS}}(\rho^{\pi_h}(s,a), \rho^{\pi_E}(s,a)).$$

Before proceeding with our main result, we need a crucial definition.

**Definition 1 (Mismatched and Matched State-action Pair)** *The state-action pair* $(s_t, h(s_t))$ *induced by the learned policy mismatches the expert demonstration* $(s_t, a_t)$, *if* $\|h(s_t) - a_t\|_2 \geq C\|\boldsymbol{\Sigma}\|_2$ *for any* $C > 0$. *Otherwise,* $(s_t, h(s_t))$ *matches the expert.*

We utilize an event

$$\Xi = \{(s_t, h(s_t)) : \|h(s_t) - a_t\|_2 \geq C\|\boldsymbol{\Sigma}\|_2 \text{ for any } C > 0\}$$

to characterize the mismatching. A descriptive example of mismatching and matching is shown in Appendix A.3.

Now we present the following theorems on the probability of exploding gradients in DE-GAIL.

**Theorem 1** *Let* $\pi_h(\cdot|s)$ *be the Gaussian stochastic policy with mean* $h(s)$ *and covariance* $\boldsymbol{\Sigma}$. *When the discriminator is set to be optimal* $D(s,a)$ *in Eq. (6), the gradient estimator of the policy loss with respect to the policy's parameter* $h$ *satisfies*

$$\|\hat{\nabla}_h D_{\mathrm{JS}}(\rho^{\pi_h}, \rho^{\pi_E})\|_2 \to \infty$$

*with a probability of* $\Pr(\|\boldsymbol{\Sigma}^{-1}(a_t - h(s_t))\|_2 \geq C \text{ for any } C > 0)$ *as* $\boldsymbol{\Sigma} \to \boldsymbol{0}$ *where*

$$\hat{\nabla}_h D_{\mathrm{JS}}(\rho^{\pi_h}, \rho^{\pi_E}) = \frac{d^{\pi_h}(s_t)\nabla_h \pi_h(a_t|s_t)}{2d^{\pi_E}(s_t)\pi_E(a_t|s_t)} \log \frac{2d^{\pi_h}(s_t)\pi_h(a_t|s_t)}{d^{\pi_h}(s_t)\pi_h(a_t|s_t) + d^{\pi_E}(s_t)\pi_E(a_t|s_t)},$$

*and* $\nabla_h \pi_h(a|s) = \pi_h(a|s)\kappa(s, \cdot)\boldsymbol{\Sigma}^{-1}(a - h(s))$.

*Proof.* See Appendix A.4. □

**Remark 1** *When* $\mathbf{\Sigma} \to \mathbf{0}$*, in other words, the policy is deterministic, we have*

$$\Pr(\|\mathbf{\Sigma}^{-1}(a_t - h(s_t))\|_2 \geq C \text{ for any } C > 0) \geq \Pr(\|a_t - h(s_t)\|_2 \geq C\|\mathbf{\Sigma}\|_2 \text{ for any } C > 0)$$
$$= \Pr(\Xi).$$

*The probability of mismatching* $\Pr(\Xi)$ *is nontrivial since* $\mathbf{\Sigma} \to \mathbf{0}$*. Theorem 1 implies that when the discriminator is set to be optimal, DE-GAIL will suffer from exploding gradients with the probabilistic lower bound* $\Pr(\Xi)$*.*

*In contrast, for a Gaussian stochastic policy (fixed* $\mathbf{\Sigma}$*), we have that* $\|\hat{\nabla}_h D_{\mathrm{JS}}(\rho^{\pi_h}, \rho^{\pi_{\mathrm{E}}})\|_2$ *is bounded referring to the proof strategy of Theorem 1. Thus, when the discriminator is set to be optimal, the Gaussian stochastic policy in GAIL will not suffer from exploding gradients. Analogous conclusions can be drawn for non-Gaussian stochastic policies.*

Theorem 1 reveals that the policy loss possibly suffers from exploding gradients when the discriminator is set to be optimal, subsequently, we will present a more universal result on the regular discriminator, which is defined as

$$\tilde{D}(s_t, a_t) = \frac{(1 + \epsilon)\rho^{\pi_E}(s_t, a_t)}{(1 + \epsilon)\rho^{\pi_E}(s_t, a_t) + (1 - \epsilon)\rho^{\pi}(s_t, a_t)}, \tag{7}$$

where $\epsilon \in (-1, 1)$. Note that

- $\tilde{D}(s_t, a_t)$ is monotonically increasing from 0 to 1 as $\epsilon \in (-1, 1)$.
- "Regular" suggests that $\tilde{D}(s_t, a_t)$ ranges in $(0, 1)$ stemmed from Eq. (2).
- $\tilde{D}(s_t, a_t)$ reaches its optimal when $\epsilon = 0$.

We next state the exploding gradients on $\tilde{D}(s_t, a_t)$.

**Theorem 2 (Main Result)** *Let* $\pi_h(\cdot|s)$ *be the Gaussian stochastic policy with mean* $h(s)$ *and covariance* $\mathbf{\Sigma}$*. When the discriminator is set to be regular* $\tilde{D}(s, a)$ *in Eq. (7), i.e.,* $\tilde{D}(s, a) \in (0, 1)$*, the gradient estimator of the policy loss with respect to the policy's parameter* $h$ *satisfies*

$$\left\| \hat{\nabla}_h \left( \mathbb{E}_{(s,a) \sim \mathcal{D}_{\mathrm{E}}}[\log(\tilde{D}(s, a))] + \mathbb{E}_{(s,a) \sim \mathcal{D}_{\mathrm{I}}}[\log(1 - \tilde{D}(s, a))] \right) \right\|_2 \to \infty$$

*with a probability of* $\Pr(\|\mathbf{\Sigma}^{-1}(a_t - h(s_t))\|_2 \geq C \text{ for any } C > 0)$ *as* $\mathbf{\Sigma} \to \mathbf{0}$*, where* $\mathcal{D}_{\mathrm{E}}$ *and* $\mathcal{D}_{\mathrm{I}}$ *denote the expert demonstration and the replay buffer of* $\pi_h$ *respectively,*

$$\hat{\nabla}_h \left( \mathbb{E}_{(s,a) \sim \mathcal{D}_{\mathrm{E}}}[\log(\tilde{D}(s, a))] + \mathbb{E}_{(s,a) \sim \mathcal{D}_{\mathrm{I}}}[\log(1 - \tilde{D}(s, a))] \right)$$
$$= \frac{d^{\pi_h}(s_t)\nabla_h \pi_h(a_t|s_t)}{d^{\pi_{\mathrm{E}}}(s_t)\pi_{\mathrm{E}}(a_t|s_t)} \log \frac{(1 - \epsilon)d^{\pi_h}(s_t)\pi_h(a_t|s_t)}{(1 + \epsilon)d^{\pi_{\mathrm{E}}}(s_t)\pi_{\mathrm{E}}(a_t|s_t) + (1 - \epsilon)d^{\pi_h}(s_t)\pi_h(a_t|s_t)}$$
$$+ \frac{2\epsilon d^{\pi_h}(s_t)\nabla_h \pi_h(a_t|s_t)}{\rho^{\pi_{\mathrm{E}}}(s_t, a_t)(1 + \epsilon) + \rho^{\pi_h}(s_t, a_t)(1 - \epsilon)},$$

*and* $\nabla_h \pi_h(a|s) = \pi_h(a|s)\kappa(s, \cdot)\mathbf{\Sigma}^{-1}(a - h(s))$.

*Proof.* See Appendix A.5. □

Analogous to Theorem 1, Theorem 2 implies

- When the discriminator ranges in $(0, 1)$, DE-GAIL will also be at risk of exploding gradients.
- When the policy loss suffers from exploding gradients during many runs, the discriminator in DE-GAIL degenerates to 0 or 1.

Differently, ST-GAIL will not suffer from exploding gradients when the discriminator ranges in $(0, 1)$.

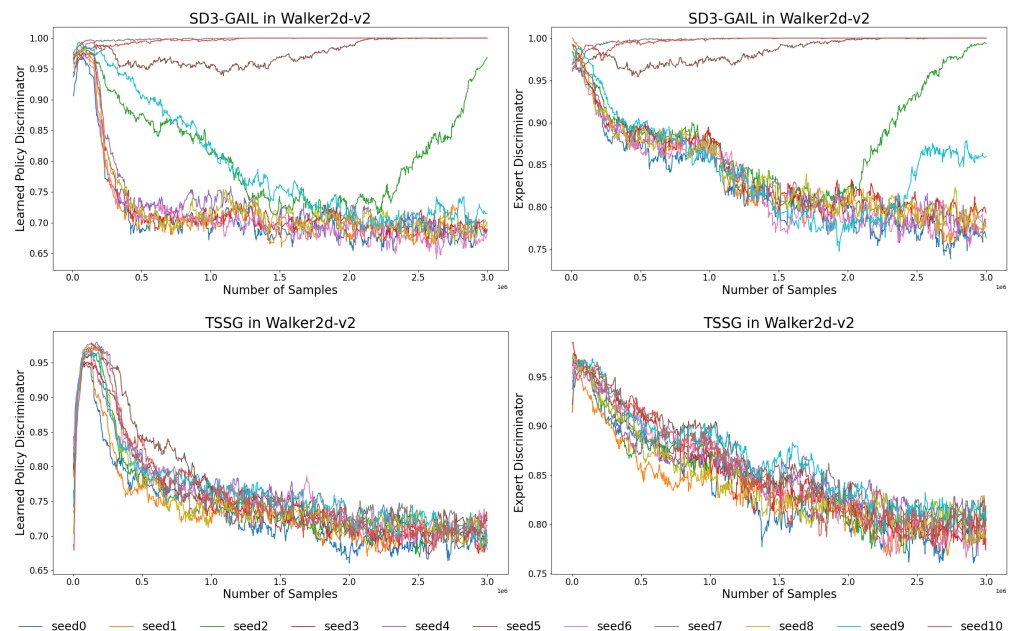

Figure 5: The discriminators of SD3-GAIL and TSSG in Walker2d-v2 under 11 seeds.

Experimental results in Fig. 5 and Fig. 6 support our theoretical analysis. Seeds 5, 7, 10 exhibit exploding gradient performances (left figure in Fig. 6) and degenerating discriminator behaviors (first row in Fig. 5), which are consistent with the invalid cases ($r \to 0$) in Fig. 4(a). Note that the gradients of TSSG and the valid cases in SD3-GAIL are in the same order

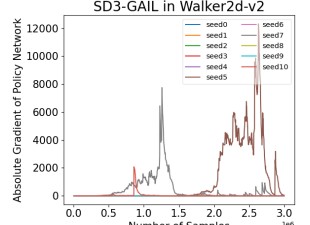
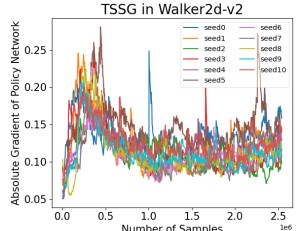

Figure 6: The absolute gradients of SD3-GAIL and TSSG policy networks in Walker2d-v2.

of magnitude, far less than the invalid cases in SD3-GAIL.

## 4.3 RELIEVING EXPLODING GRADIENTS WITH REWARD MODIFICATION

Kostrikov et al. (2019) introduced the reward function $r_2(s_t, a_t) = \log(D(s_t, a_t)) - \log(1 - D(s_t, a_t))$ of AIRL into GAIL, and illustrated a reduction in instability through comparative experiments against GAIL with PLR $r_1(s_t, a_t) = -\log(1 - D(s_t, a_t))$. The reward function of AIRL is defined as the *combination reward function* (CR) in Wang & Li (2021). For written convenience, DE-GAIL with PLR and CR are called PLR-DE-GAIL and CR-DE-GAIL respectively.

Further, in this section, we proceed with this reduction of instability from a theoretical explanation, in other words, whether CR-DE-GAIL permits a relief in exploding gradient probability over PLR-DE-GAIL.

Theorem 1 points out that the mismatched state-action pair results in exploding gradients. Further, what truly pertains to our discussion in the sequel is the behavior of the discriminator. This is due to its unified characteristic that lies in the interval $[0, 1]$. Now we present the following Lemma on the mismatching in the view of the discriminator.

**Proposition 1** *When the discriminator is set to be optimal $D^*(s, a)$ in Eq. (6), we have*

$$D^*(s_t, a_t) \approx 1 \Leftrightarrow h(s_t) \text{ mismatches } a_t.$$

*Proof.* See Appendix A.6. □

Lemma 1 indicates that exploding gradients depend on the distance between the discriminator's value and 1, or they depend on the degree of $r_i(s_t, a_t)$ for $i = 1, 2$ when $r$ approaches infinity. This is due to the monotonicity of both $r_1(s_t, a_t)$ and $r_2(s_t, a_t)$. we thus obtain

$$r_1(s_t, a_t) \approx \infty \text{ and } r_2(s_t, a_t) \approx \infty \tag{8}$$

as $D(s_t, a_t) \approx 1$.

Naturally, to prevent exploding gradients, we make the constraints that $r_i(s_t, a_t) \leq C$, $i = 1, 2$, for some appropriate constant $C$. In contrast, the outlier of the discriminator can be characterized as $r_i(s_t, a_t) > C$ for $i = 1, 2$, which is referred to as the following.

**Definition 2** *When the discriminator is set to be optimal $D^*(s, a)$ in Eq. (6), the outliers of the discriminator are defined in $[\alpha, 1]$ such that $r_1(s_t, a_t) \geq C$. Similarly, under the same upper bound $C$, the outliers of the discriminator are defined in $[\beta, 1]$ for $r_2(s_t, a_t)$.*

We note that the training process will suffer from exploding gradients when the discriminator comes to rest in $[\alpha, 1]$. The remission of exploding gradients in CR-DE-GAIL is presented in the following proposition.

**Proposition 2** *When the discriminator is set to be optimal $D^*(s, a)$ in Eq. (6), we have $\beta \geq \alpha$.*

*Proof.* See Appendix A.7. □

Proposition 2 reveals that the discriminator in CR-DE-GAIL exhibits a smaller interval of outliers than that in PLR-DE-GAIL, decreasing the probability of gradient explosion. Our conclusion is consistent with the claim in Kostrikov et al. (2019) (Fig. 5).

## 5 CONCLUSION

In this paper, we have explored the principle of instability in DE-GAIL. We first experimentally show the extreme instability performance of DE-GAIL algorithms compared to ST-GAIL. Subsequently, our proof manifests that the gradient of the deterministic policy loss with respect to the policy will suffer from exploding with some probability, thereby leading to training failure. In comparison, we present the compatibility between stochastic policy and GAIL. Finally, the modified reward function is shown to remedy the exploding gradients.

Reducing the probability of exploding gradients is under consideration. By introducing the idea of clipping the reward into SD3-GAIL, we have discovered some interesting phenomena. For a preliminary verification of our algorithm, please refer to Appendix A.8. Specifically, the fact that clipping the reward that leads to the outlier of the discriminator shows its superiority in the stability of DE-GAIL. Further analysis that improves the sample efficiency while keeping low exploding gradient probability is left for future works.

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

## A  APPENDIX

### A.1  DE-GAIL ALGORITHM

---
**Algorithm 1** GAIL with deterministic policy algorithms

---
1: **Input:** Expert demonstrations, initialize the learned policy $\pi_\theta$ and discriminator network $D$, some RL deterministic policy algorithms, such as SD3, TD3 and DDPG.
2: **for** iteration $0, 1, 2, \cdots$ **do**
3:    Update $D$ by maximizing $\mathbb{E}_{(s,a)\sim\rho^{\pi_E}}[\log(D(s,a))] + \mathbb{E}_{(s,a)\sim\rho^\pi}[\log(1 - D(s,a))]$.
4:    Calculate PLR $r(s,a) = -\log(1 - D(s,a))$.
5:    Update $\theta$ by a deterministic policy algorithm.
6: **end for**

---

## A.2 THE INVALIDITY IN DE-GAIL

The instability of DE-GAIL is mainly imputed to the invalid case, i.e., the reward function approaches zero. To illustrate this invalidity, consider a specific MDP $\mathcal{M}$: $\mathcal{S}$ contains $(g+1)$ states $s_0, s_1, ..., s_g$, where $s_g$ is the terminal state. The action $a_{i \to j}$ in $\mathcal{A}$ is such that $s_j \sim p(\cdot|s_i, a_{i \to j})$ satisfies that $|i - j| \leq g/4$ for $i \neq j$. The expert demonstration is shown in Fig. 7.

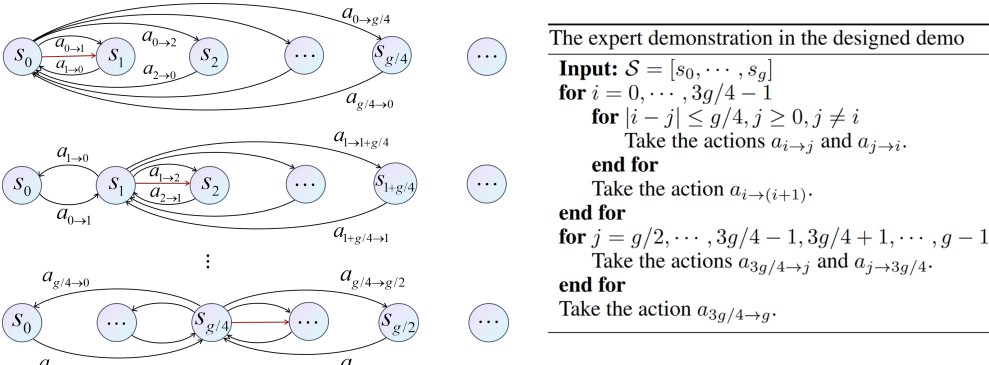

The expert demonstration in the designed demo

**Input:** $\mathcal{S} = [s_0, \cdots, s_g]$
**for** $i = 0, \cdots, 3g/4 - 1$
    **for** $|i - j| \leq g/4, j \geq 0, j \neq i$
        Take the actions $a_{i \to j}$ and $a_{j \to i}$.
    **end for**
    Take the action $a_{i \to (i+1)}$.
**end for**
**for** $j = g/2, \cdots, 3g/4 - 1, 3g/4 + 1, \cdots, g - 1$
    Take the actions $a_{3g/4 \to j}$ and $a_{j \to 3g/4}$.
**end for**
Take the action $a_{3g/4 \to g}$.

Figure 7: Expert demonstration. Each state transfers to the neighboring $g/4$ states and back to itself from left to right. **Left:** The schematic of the expert demonstration, where the red arrow is the last action taken in each state. **Right:** The pseudocode of the expert demonstration.

For the state $s_t$ ($0 \leq t < 3g/4$), the action $a_{t \to t+1}$ occurs 3 times while others occur at most twice in the expert demonstration. Note that

$$D^*(s, a) = \frac{\rho^{\pi_E}(s, a)}{\rho^{\pi_E}(s, a) + \rho^{\pi}(s, a)} = \frac{1}{1 + \frac{\rho^{\pi}(s,a)}{\rho^{\pi_E}(s,a)}},$$

$D^*(s, a)$ is monotonically increasing with respect to $\rho^{\pi_E}(s, a)$. Thus, the more frequent expert state-action pair tends to attain a higher value of the discriminator, thereby resulting in higher PLR. Meanwhile, policy training in RL aims to maximize the expected reward-to-go. Therefore, the most frequent action $a_{t \to t+1}$ is the best choice for a deterministic learned policy under each $s_t$. The detailed trajectory is shown in Fig. 8.

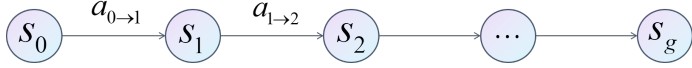

Figure 8: The trajectory of the deterministic learned policy. From $s_0$ to the terminal state $s_g$ sequentially.

We now show the behavior of rewards for state-action pairs in the deterministic policy.

**Proposition 3** *In the MDP $\mathcal{M}$, when the discriminator is set to be optimal, PLR of $(s_t, a_{t \to t+1})$ satisfies*

$$r(s_t, a_{t \to t+1}) = -\log(1 - D(s_t, a_{t \to t+1})) \to 0 \quad as \quad |\mathcal{S}| \to \infty,$$

*where*

$$D(s_t, a_{t \to t+1}) = \begin{cases} \frac{48g}{11g^2 + 72g - 16}, & 0 \leq t < \frac{3g}{4}, \\ \frac{16g}{11g^2 + 40g - 16}, & t = \frac{3g}{4}, \\ 0, & \text{otherwise.} \end{cases}$$

*Proof.* The length of the expert trajectory is

$$N = 2\left(\frac{g}{4} + (\frac{g}{4} + 1) + \cdots + (\frac{g}{4} + \frac{g}{4} - 1)\right) + 2(\frac{g}{2} + 1)\frac{g}{2} - 1 + \frac{3g}{4}$$

$$= \frac{11g^2 + 24g - 16}{16}.$$

Note that the first term $2(g/4 + (g/4 + 1) + \cdots + (g/4 + g/4 - 1))$ comes from executing actions $a_{i \to j}, a_{j \to i}$ in states $s_0, s_1, \cdots, s_{g/4-1}$, the second term $2(g/2 + 1)g/2 - 1$ is derived by executing actions $a_{i \to j}, a_{j \to i}$ in states $s_{g/4}, s_{g/4+1}, \cdots, s_{3g/4}$, and the last term $3g/4$ is obtained by executing the action $a_{i \to i+1}$.

Denote the expert policy and the deterministic policy as $\pi_E$ and $h$, respectively. For the $(s_t, a_{t \to t+1})$, $0 \le t < 3g/4$ in the trajectory of the deterministic policy, the optimal discriminator is (Kostrikov et al., 2019):

$$D(s_t, a_{t \to t+1}) = \frac{\rho^{\pi_E}(s_t, a_{t \to t+1})}{\rho^{\pi_E}(s_t, a_{t \to t+1}) + \rho^h(s_t, a_{t \to t+1})}$$

$$= \frac{\frac{3}{N}}{\frac{3}{N} + \frac{1}{g}}$$

$$= \frac{48g}{11g^2 + 72g - 16}.$$

For $t = 3g/4$, we have

$$D(s_{3g/4}, a_{3g/4 \to 3g/4+1}) = \frac{16g}{11g^2 + 40g - 16}.$$

While for any $t > 3g/4$, the value $D(s_t, a_{t \to t+1})$ is zero since it never appears in the expert demonstrations. To sum up, when the discriminator achieves its optimal,

$$r(s_t, a_{t \to t+1}) = -\log(1 - D((s_t, a_{t \to t+1}))) \to 0 \quad as \quad |\mathcal{S}| \to \infty.$$

$\square$

**Remark 2** *Note that under the deterministic policy, PLR of the best action approaches 0. As a result, we have $r \to 0$ for all state-action pairs.*

*In contrast, for a stochastic policy reaching the expert level, the length of the expert trajectory and the stochastic policy have the same order $\mathcal{O}(|\mathcal{S}|^2)$. Here $|\mathcal{S}|$ denotes the size of $\mathcal{S}$. Referring to the proof strategy, PLR will not decline to zero.*

Proposition 3 illustrates the invalid case, implying the instability in DE-GAIL.

## A.3 An Descriptive Example of Mismatching and Matching

The probability density contour map of the expert action demonstration in state $s_t$ is shown in Fig. 9.

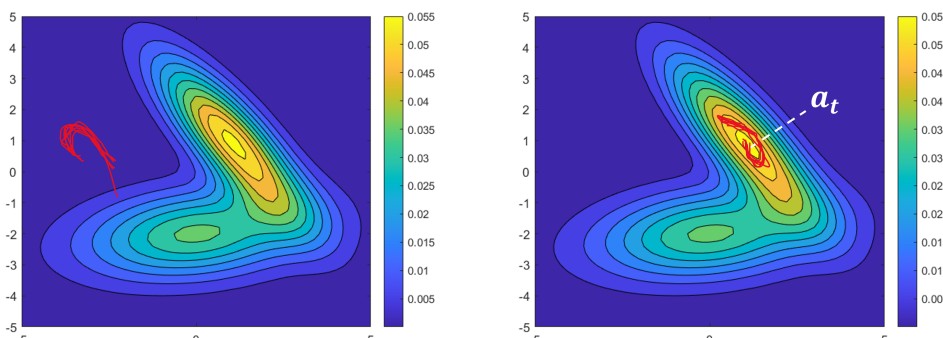

Figure 9: The process of the mismatched case and the matched case on a descriptive example of a two-dimensional action space (x-axis and y-axis). **Left:** The mismatched case. **Right:** The matched case. Without loss of generality, the threshold of matching is set as 0.035. The trajectories used to train the learned policies are shown in red curves. The matching is that $h(s_t)$ lies in the neighborhood of $a_t$, which has a high probabilistic density in the expert demonstration.

## A.4 Proof of Theorem 1

**Theorem 1** *Let $\pi_h(\cdot|s)$ be the Gaussian stochastic policy with mean $h(s)$ and covariance $\Sigma$. When the discriminator is set to be optimal $D(s,a)$ in Eq. (6), the gradient estimator of the policy loss with respect to the policy's parameter $h$ satisfies*

$$\|\hat{\nabla}_h D_{\mathrm{JS}}(\rho^{\pi_h}, \rho^{\pi_{\mathrm{E}}})\|_2 \to \infty$$

*with a probability of $\mathrm{Pr}(\|\Sigma^{-1}(a_t - h(s_t))\|_2 \geq C$ for any $C > 0)$ as $\Sigma \to \mathbf{0}$ where*

$$\hat{\nabla}_h D_{\mathrm{JS}}(\rho^{\pi_h}, \rho^{\pi_{\mathrm{E}}}) = \frac{d^{\pi_h}(s_t)\nabla_h \pi_h(a_t|s_t)}{2d^{\pi_{\mathrm{E}}}(s_t)\pi_{\mathrm{E}}(a_t|s_t)} \log \frac{2d^{\pi_h}(s_t)\pi_h(a_t|s_t)}{d^{\pi_h}(s_t)\pi_h(a_t|s_t) + d^{\pi_{\mathrm{E}}}(s_t)\pi_{\mathrm{E}}(a_t|s_t)},$$

*and $\nabla_h \pi_h(a|s) = \pi_h(a|s)\kappa(s, \cdot)\Sigma^{-1}(a - h(s))$.*

*Proof.* Through importance sampling which transfers the learned state-action distribution to the expert demonstration distribution, the JS divergence can be rewritten from the definition in Eq. (4) as

$$
\begin{aligned}
&D_{\mathrm{JS}}(\rho^{\pi_h}, \rho^{\pi_{\mathrm{E}}}) \\
&= \frac{1}{2}D_{\mathrm{KL}}\left(\rho^{\pi_h}, \frac{\rho^{\pi_h} + \rho^{\pi_{\mathrm{E}}}}{2}\right) + \frac{1}{2}D_{\mathrm{KL}}\left(\rho^{\pi_{\mathrm{E}}}, \frac{\rho^{\pi_h} + \rho^{\pi_{\mathrm{E}}}}{2}\right) \\
&= \frac{1}{2}\mathbb{E}_{(s,a)\sim\mathcal{D}_{\mathrm{I}}}\left[\log \frac{2\rho^{\pi_h}(s,a)}{\rho^{\pi_h}(s,a) + \rho^{\pi_{\mathrm{E}}}(s,a)}\right] + \frac{1}{2}\mathbb{E}_{(s,a)\sim\mathcal{D}_{\mathrm{E}}}\left[\log \frac{2\rho^{\pi_{\mathrm{E}}}(s,a)}{\rho^{\pi_h}(s,a) + \rho^{\pi_{\mathrm{E}}}(s,a)}\right] \\
&= \frac{1}{2}\mathbb{E}_{(s,a)\sim\mathcal{D}_{\mathrm{E}}}\left[\frac{\rho^{\pi_h}(s,a)}{\rho^{\pi_{\mathrm{E}}}(s,a)} \log \frac{2\rho^{\pi_h}(s,a)}{\rho^{\pi_h}(s,a) + \rho^{\pi_{\mathrm{E}}}(s,a)} + \log \frac{2\rho^{\pi_{\mathrm{E}}}(s,a)}{\rho^{\pi_h}(s,a) + \rho^{\pi_{\mathrm{E}}}(s,a)}\right],
\end{aligned}
\tag{9}
$$

where $\mathcal{D}_{\mathrm{E}}$ and $\mathcal{D}_{\mathrm{I}}$ denote the expert demonstration and the replay buffer of $\pi_h$ respectively. Then we can approximate the gradient of Eq. (9) with respect to $h$ with

$$
\begin{aligned}
&\hat{\nabla}_h D_{\mathrm{JS}}(\rho^{\pi_h}, \rho^{\pi_{\mathrm{E}}}) \\
&\overset{(i)}{=} \frac{1}{2} \nabla_h \left( \frac{\rho^{\pi_h}(s_t, a_t)}{\rho^{\pi_{\mathrm{E}}}(s_t, a_t)} \log \frac{2\rho^{\pi_h}(s_t, a_t)}{\rho^{\pi_h}(s_t, a_t) + \rho^{\pi_{\mathrm{E}}}(s_t, a_t)} + \log \frac{2\rho^{\pi_{\mathrm{E}}}(s_t, a_t)}{\rho^{\pi_h}(s_t, a_t) + \rho^{\pi_{\mathrm{E}}}(s_t, a_t)} \right) \\
&\overset{(ii)}{=} \frac{1}{2} \left( \frac{d^{\pi_h}(s_t) \nabla_h \pi_h(a_t|s_t)}{d^{\pi_{\mathrm{E}}}(s_t) \pi_{\mathrm{E}}(a_t|s_t)} \log \frac{2d^{\pi_h}(s_t) \pi_h(a_t|s_t)}{d^{\pi_h}(s_t) \pi_h(a_t|s_t) + d^{\pi_{\mathrm{E}}}(s_t) \pi_{\mathrm{E}}(a_t|s_t)} \right. \\
&\quad + \frac{\rho^{\pi_h}(s_t, a_t)}{\rho^{\pi_{\mathrm{E}}}(s_t, a_t)} \cdot \frac{\rho^{\pi_h}(s_t, a_t) + \rho^{\pi_{\mathrm{E}}}(s_t, a_t)}{2\rho^{\pi_h}(s_t, a_t)} \\
&\quad \cdot \frac{2d^{\pi_h}(s_t) \nabla_h \pi_h(a_t|s_t) \left( \rho^{\pi_h}(s_t, a_t) + \rho^{\pi_{\mathrm{E}}}(s_t, a_t) \right) - 2\rho^{\pi_h}(s_t, a_t) d^{\pi_h}(s_t) \nabla_h \pi_h(a_t|s_t)}{\left( \rho^{\pi_h}(s_t, a_t) + \rho^{\pi_{\mathrm{E}}}(s_t, a_t) \right)^2} \\
&\quad \left. - \frac{\rho^{\pi_h}(s_t, a_t) + \rho^{\pi_{\mathrm{E}}}(s_t, a_t)}{2\rho^{\pi_{\mathrm{E}}}(s_t, a_t)} \cdot \frac{2\rho^{\pi_{\mathrm{E}}}(s_t, a_t) d^{\pi_h}(s_t) \nabla_h \pi_h(a_t|s_t)}{\left( \rho^{\pi_h}(s_t, a_t) + \rho^{\pi_{\mathrm{E}}}(s_t, a_t) \right)^2} \right) \\
&\overset{(iii)}{=} \frac{1}{2} \left( \frac{d^{\pi_h}(s_t) \nabla_h \pi_h(a_t|s_t)}{d^{\pi_{\mathrm{E}}}(s_t) \pi_{\mathrm{E}}(a_t|s_t)} \log \frac{2d^{\pi_h}(s_t) \pi_h(a_t|s_t)}{d^{\pi_h}(s_t) \pi_h(a_t|s_t) + d^{\pi_{\mathrm{E}}}(s_t) \pi_{\mathrm{E}}(a_t|s_t)} \right. \\
&\quad \left. + \frac{d^{\pi_h}(s_t) \nabla_h \pi_h(a_t|s_t)}{\rho^{\pi_h}(s_t, a_t) + \rho^{\pi_{\mathrm{E}}}(s_t, a_t)} - \frac{d^{\pi_h}(s_t) \nabla_h \pi_h(a_t|s_t)}{\rho^{\pi_h}(s_t, a_t) + \rho^{\pi_{\mathrm{E}}}(s_t, a_t)} \right) \\
&\overset{(iv)}{=} \frac{d^{\pi_h}(s_t) \nabla_h \pi_h(a_t|s_t)}{2d^{\pi_{\mathrm{E}}}(s_t) \pi_{\mathrm{E}}(a_t|s_t)} \log \frac{2d^{\pi_h}(s_t) \pi_h(a_t|s_t)}{d^{\pi_h}(s_t) \pi_h(a_t|s_t) + d^{\pi_{\mathrm{E}}}(s_t) \pi_{\mathrm{E}}(a_t|s_t)},
\end{aligned}
\tag{10}
$$

where (ii) comes from Eq. (1). By the fact that

$$
\nabla_h \pi_h(a|s) = \pi_h(a|s) \nabla_h \log \pi_h(a|s) = \pi_h(a|s) \kappa(s, \cdot) \boldsymbol{\Sigma}^{-1}(a - h(s)),
\tag{11}
$$

Eq. (10) can be shown that

$$
\begin{aligned}
&\| \hat{\nabla}_h D_{\mathrm{JS}}(\rho^{\pi_h}, \rho^{\pi_{\mathrm{E}}}) \|_2 \\
&= \left\| \frac{d^{\pi_h}(s_t) \pi_h(a_t|s_t) \kappa(s_t, \cdot) \boldsymbol{\Sigma}^{-1}(a_t - h(s_t))}{2d^{\pi_{\mathrm{E}}}(s_t) \pi_{\mathrm{E}}(a_t|s_t)} \log \frac{2d^{\pi_h}(s_t) \pi_h(a_t|s_t)}{d^{\pi_h}(s_t) \pi_h(a_t|s_t) + d^{\pi_{\mathrm{E}}}(s_t) \pi_{\mathrm{E}}(a_t|s_t)} \right\|_2.
\end{aligned}
$$

Then it follows that $\| \hat{\nabla}_h D_{\mathrm{JS}}(\rho^{\pi_h}, \rho^{\pi_{\mathrm{E}}}) \|_2 \to \infty$ with a probability of $\mathrm{Pr}(\| \boldsymbol{\Sigma}^{-1}(a_t - h(s_t)) \|_2 \geq C$ for any $C > 0)$ as $\boldsymbol{\Sigma} \to \mathbf{0}$. $\qquad \square$

## A.5  PROOF OF THEOREM 2

**Theorem 2 (Main Result)** *Let $\pi_h(\cdot|s)$ be the Gaussian stochastic policy with mean $h(s)$ and co-variance $\boldsymbol{\Sigma}$. When the discriminator is set to be regular $\tilde{D}(s, a)$ in Eq. (7), i.e., $\tilde{D}(s, a) \in (0, 1)$, the gradient estimator of the policy loss with respect to the policy's parameter $h$ satisfies*

$$
\left\| \hat{\nabla}_h \left( \mathbb{E}_{(s,a) \sim \mathcal{D}_{\mathrm{E}}} [\log(\tilde{D}(s, a))] + \mathbb{E}_{(s,a) \sim \mathcal{D}_{\mathrm{I}}} [\log(1 - \tilde{D}(s, a))] \right) \right\|_2 \to \infty
$$

*with a probability of $\mathrm{Pr}(\| \boldsymbol{\Sigma}^{-1}(a_t - h(s_t)) \|_2 \geq C$ for any $C > 0)$ as $\boldsymbol{\Sigma} \to \mathbf{0}$, where $\mathcal{D}_{\mathrm{E}}$ and $\mathcal{D}_{\mathrm{I}}$ denote the expert demonstration and the replay buffer of $\pi_h$ respectively,*

$$
\begin{aligned}
&\hat{\nabla}_h \left( \mathbb{E}_{(s,a) \sim \mathcal{D}_{\mathrm{E}}} [\log(\tilde{D}(s, a))] + \mathbb{E}_{(s,a) \sim \mathcal{D}_{\mathrm{I}}} [\log(1 - \tilde{D}(s, a))] \right) \\
&= \frac{d^{\pi_h}(s_t) \nabla_h \pi_h(a_t|s_t)}{d^{\pi_{\mathrm{E}}}(s_t) \pi_{\mathrm{E}}(a_t|s_t)} \log \frac{(1 - \epsilon) d^{\pi_h}(s_t) \pi_h(a_t|s_t)}{(1 + \epsilon) d^{\pi_{\mathrm{E}}}(s_t) \pi_{\mathrm{E}}(a_t|s_t) + (1 - \epsilon) d^{\pi_h}(s_t) \pi_h(a_t|s_t)} \\
&\quad + \frac{2\epsilon d^{\pi_h}(s_t) \nabla_h \pi_h(a_t|s_t)}{\rho^{\pi_{\mathrm{E}}}(s_t, a_t)(1 + \epsilon) + \rho^{\pi_h}(s_t, a_t)(1 - \epsilon)},
\end{aligned}
$$

*and $\nabla_h \pi_h(a|s) = \pi_h(a|s) \kappa(s, \cdot) \boldsymbol{\Sigma}^{-1}(a - h(s))$.*

*Proof.* Referring to the proof strategy of Theorem 1, the learned state-action distribution can be transferred to the expert demonstration distribution by importance sampling. Thus when the discriminator is set to be regular $\tilde{D}(s, a)$, we can write the policy objective from the optimization

problem in Eq. (2) as

$$\mathbb{E}_{(s,a)\sim\mathcal{D}_{\mathrm{E}}}[\log(\tilde{D}(s,a))] + \mathbb{E}_{\mathcal{D}_{\mathrm{I}}}[\log(1 - \tilde{D}(s,a))]$$

$$= \mathbb{E}_{(s,a)\sim\mathcal{D}_{\mathrm{E}}}\left[\log\frac{\rho^{\pi_{\mathrm{E}}}(s,a)(1+\epsilon)}{\rho^{\pi_{\mathrm{E}}}(s,a)(1+\epsilon) + \rho^{\pi_h}(s,a)(1-\epsilon)}\right]$$

$$+ \mathbb{E}_{(s,a)\sim\mathcal{D}_{\mathrm{I}}}\left[\log\frac{\rho^{\pi_h}(s,a)(1-\epsilon)}{\rho^{\pi_{\mathrm{E}}}(s,a)(1+\epsilon) + \rho^{\pi_h}(s,a)(1-\epsilon)}\right]$$

$$= \mathbb{E}_{(s,a)\sim\mathcal{D}_{\mathrm{E}}}\left[\log\frac{\rho^{\pi_{\mathrm{E}}}(s,a)(1+\epsilon)}{\rho^{\pi_{\mathrm{E}}}(s,a)(1+\epsilon) + \rho^{\pi_h}(s,a)(1-\epsilon)}\right.$$

$$\left. + \frac{\rho^{\pi_h}(s,a)}{\rho^{\pi_{\mathrm{E}}}(s,a)}\log\frac{\rho^{\pi_h}(s,a)(1-\epsilon)}{\rho^{\pi_{\mathrm{E}}}(s,a)(1+\epsilon) + \rho^{\pi_h}(s,a)(1-\epsilon)}\right]. \quad (12)$$

Then the gradient of Eq. (12) can be approximated with

$$\hat{\nabla}_h\left(\mathbb{E}_{(s,a)\sim\mathcal{D}_{\mathrm{E}}}[\log(\tilde{D}(s,a))] + \mathbb{E}_{\mathcal{D}_{\mathrm{I}}}[\log(1 - \tilde{D}(s,a))]\right)$$

$$= \nabla_h\left(\log\frac{\rho^{\pi_{\mathrm{E}}}(s_t,a_t)(1+\epsilon)}{\rho^{\pi_{\mathrm{E}}}(s_t,a_t)(1+\epsilon) + \rho^{\pi_h}(s_t,a_t)(1-\epsilon)}\right.$$

$$\left. + \frac{\rho^{\pi_h}(s_t,a_t)}{\rho^{\pi_{\mathrm{E}}}(s_t,a_t)}\log\frac{\rho^{\pi_h}(s_t,a_t)(1-\epsilon)}{\rho^{\pi_{\mathrm{E}}}(s_t,a_t)(1+\epsilon) + \rho^{\pi_h}(s_t,a_t)(1-\epsilon)}\right)$$

$$= -\frac{\rho^{\pi_{\mathrm{E}}}(s_t,a_t)(1+\epsilon) + \rho^{\pi_h}(s_t,a_t)(1-\epsilon)}{\rho^{\pi_{\mathrm{E}}}(s_t,a_t)(1+\epsilon)} \cdot \frac{\rho^{\pi_{\mathrm{E}}}(s_t,a_t)d^{\pi_h}(s_t)\nabla_h\pi_h(a_t|s_t)(1+\epsilon)(1-\epsilon)}{\left(\rho^{\pi_{\mathrm{E}}}(s_t,a_t)(1+\epsilon) + \rho^{\pi_h}(s_t,a_t)(1-\epsilon)\right)^2}$$

$$+ \frac{d^{\pi_h}(s_t)\nabla_h\pi_h(a_t|s_t)}{d^{\pi_{\mathrm{E}}}(s_t)\pi_{\mathrm{E}}(a_t|s_t)}\log\frac{(1-\epsilon)d^{\pi_h}(s_t)\pi_h(a_t|s_t)}{(1+\epsilon)d^{\pi_{\mathrm{E}}}(s_t)\pi_{\mathrm{E}}(a_t|s_t) + (1-\epsilon)d^{\pi_h}(s_t)\pi_h(a_t|s_t)}$$

$$+ \frac{\rho^{\pi_h}(s_t,a_t)}{\rho^{\pi_{\mathrm{E}}}(s_t,a_t)} \cdot \frac{\rho^{\pi_{\mathrm{E}}}(s_t,a_t)(1+\epsilon) + \rho^{\pi_h}(s_t,a_t)(1-\epsilon)}{\rho^{\pi_h}(s_t,a_t)(1-\epsilon)}$$

$$\cdot \frac{(1-\epsilon)(1+\epsilon)\rho^{\pi_{\mathrm{E}}}(s_t,a_t)d^{\pi_h}(s_t)\nabla_h\pi_h(a_t|s_t)}{\left(\rho^{\pi_{\mathrm{E}}}(s_t,a_t)(1+\epsilon) + \rho^{\pi_h}(s_t,a_t)(1-\epsilon)\right)^2}$$

$$= \frac{d^{\pi_h}(s_t)\nabla_h\pi_h(a_t|s_t)}{d^{\pi_{\mathrm{E}}}(s_t)\pi_{\mathrm{E}}(a_t|s_t)}\log\frac{(1-\epsilon)d^{\pi_h}(s_t)\pi_h(a_t|s_t)}{(1+\epsilon)d^{\pi_{\mathrm{E}}}(s_t)\pi_{\mathrm{E}}(a_t|s_t) + (1-\epsilon)d^{\pi_h}(s_t)\pi_h(a_t|s_t)}$$

$$- \frac{(1-\epsilon)d^{\pi_h}(s_t)\nabla_h\pi_h(a_t|s_t)}{\rho^{\pi_{\mathrm{E}}}(s_t,a_t)(1+\epsilon) + \rho^{\pi_h}(s_t,a_t)(1-\epsilon)} + \frac{(1+\epsilon)d^{\pi_h}(s_t)\nabla_h\pi_h(a_t|s_t)}{\rho^{\pi_{\mathrm{E}}}(s_t,a_t)(1+\epsilon) + \rho^{\pi_h}(s_t,a_t)(1-\epsilon)}$$

$$= \frac{d^{\pi_h}(s_t)\nabla_h\pi_h(a_t|s_t)}{d^{\pi_{\mathrm{E}}}(s_t)\pi_{\mathrm{E}}(a_t|s_t)}\log\frac{(1-\epsilon)d^{\pi_h}(s_t)\pi_h(a_t|s_t)}{(1+\epsilon)d^{\pi_{\mathrm{E}}}(s_t)\pi_{\mathrm{E}}(a_t|s_t) + (1-\epsilon)d^{\pi_h}(s_t)\pi_h(a_t|s_t)}$$

$$+ \frac{2\epsilon d^{\pi_h}(s_t)\nabla_h\pi_h(a_t|s_t)}{\rho^{\pi_{\mathrm{E}}}(s_t,a_t)(1+\epsilon) + \rho^{\pi_h}(s_t,a_t)(1-\epsilon)}. \quad (13)$$

Plugging Eq. (11) into Eq. (13), when $\|\mathbf{\Sigma}^{-1}(a_t - h(s_t))\|_2 \geq C$ for any $C > 0$, we have

$$\left\|\hat{\nabla}_h\left(\mathbb{E}_{(s,a)\sim\mathcal{D}_{\mathrm{E}}}[\log(\tilde{D}(s,a))] + \mathbb{E}_{\mathcal{D}_{\mathrm{I}}}[\log(1 - \tilde{D}(s,a))]\right)\right\|_2 \to \infty.$$

$\square$

## A.6 PROOF OF PROPOSITION 1

**Proposition 1** *When the discriminator is set to be optimal $D^*(s,a)$ in Eq. (6), we have*

$$D^*(s_t,a_t) \approx 1 \Leftrightarrow h(s_t) \text{ mismatches } a_t.$$

*Proof.* The optimal discriminator of $(s_t, a_t)$ can be denoted by

$$D^*(s_t,a_t) = \frac{\rho^{\pi_{\mathrm{E}}}(s_t,a_t)}{\rho^{\pi_{\mathrm{E}}}(s_t,a_t) + \rho^{\pi_h}(s_t,a_t)}.$$

We can derive that the necessary and sufficient condition of $D^*(s_t,a_t) \approx 1$ is that $\rho^{\pi_h}(s_t,a_t) \approx 0$, i.e., $(s_t, h(s_t))$ mismatches $(s_t, a_t)$. $\square$

## A.7 PROOF OF PROPOSITION 2

**Proposition 2** *When the discriminator is set to be optimal $D^*(s, a)$ in Eq. (6), we have $\beta \geq \alpha$.*

*Proof.* When $r_i(s_t, a_t) = C$, $i = 1, 2$, we obtain $\log \beta - \log(1 - \beta) = -\log(1 - \alpha)$, which is followed by

$$\beta - \alpha = \frac{\alpha^2 - 2\alpha + 1}{2 - \alpha} \geq 0.$$

$\square$

## A.8 SD3-GAIL WITH CLIPPED REWARD

Clipping reward shows its superiority in the stability of DE-GAIL but at the expense of lower sample efficiency.

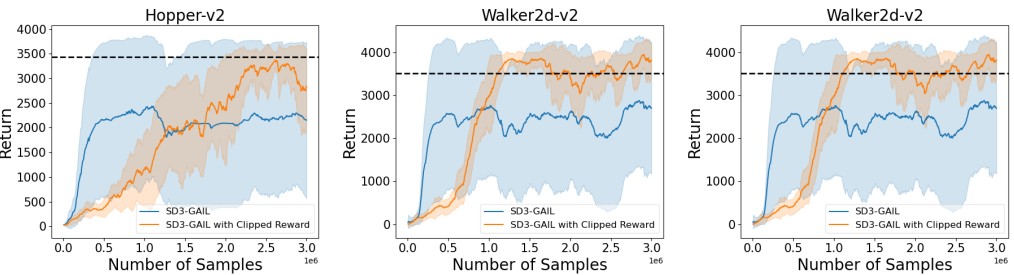

Figure 10: Comparison of SD3-GAIL and SD3-GAIL with clipped reward in three different environments.