# OpenReview forum: "Incompatibility between Deterministic Policy and Generative Adversarial Imitation Learning"
_ICLR.cc/2023/Conference — Submitted to ICLR 2023_

### Official Review · Reviewer_rV27 · 2022-10-15

**Confidence:** 3
**Correctness:** 2
**Technical Novelty And Significance:** 3
**Empirical Novelty And Significance:** 2
**Recommendation:** 3

**Clarity, Quality, Novelty And Reproducibility:**

As discussed, in its current form, I found it very difficult to follow the writing of the paper.

**Strength And Weaknesses:**

# Strenghts

* The authors try to thorougly understand existing algorithms and investigate their weaknesses, which is an interesting and value contribution.
* The authors try to support their insights using experiments and toy models, which is helpful.

# Weaknesses

In my opinion, the biggest weakness of the paper is insufficient clarity in writing. At times, wrong grammar or word-use make reading more difficult, but the intended meaning can still be inferred. Other times however, I found myself unable to follow the writing. Furthermore, the authors sometimes introduce concepts in quotations marks without ever clearly explaining their meaning (e.g. "aggressive interval", "non-confidence" or "invalidity").

My other big concern is with respect to Theorem 1 in which they show the probability of exploding gradients. As far as I understand T1, the argument is that the gradient of a JSD divergence w.r.t a deterministic policy can explode. However, I'm not srue this can explain the experimental results, as this JSD is not the optimization target - instead an RL loss is being used.

Lastly, more experimental results would be needed.

**Summary Of The Paper:**

The paper investigates the case when deterministic policies are learned in a GAN framework. The authors describe occuring instabilities and attribute them to exploding gradients.

**Summary Of The Review:**

I'm not convinced of the validity of the key results of the paper (T1). However, more importantly, I found the writing at times too difficult to follow, so I am unable to appropriately evaluate the contributions of the paper.

---

> ### Author Response · Authors · 2022-11-19
> **Answer to Reviewer rV27**
>
> 1. *In my opinion, the biggest weakness of the paper is insufficient clarity in writing. At times, wrong grammar or word-use make reading more difficult, but the intended meaning can still be inferred. Other times however, I found myself unable to follow the writing. Furthermore, the authors sometimes introduce concepts in quotations marks without ever clearly explaining their meaning (e.g. "aggressive interval", "non-confidence" or "invalidity").*
>
>  **Answer.** Thanks for your careful reading. We have checked the grammar using the software Grammarly and avoided the unclear statements.
>
> Actually, "validity" and "invalidity" stem from the phenomenon in experiments. Specifically, learning effects under some seeds are capable of reaching expert levels (valid), while others hardly learn anything (invalid); see paragraph 2, page 2 in the revised version.
>
> In addition, we rewrite the unclear concepts. "Aggressive interval" is only an intutive description. In fact, we should use the common statements "outlier" for preciseness; see Section 4.3 in the revised version. "Non-confident" in the original version is a interpretive description in the sense of low sample efficiency, rather than strictly definition. In this revision, we omit this misunderstanding statement. Thanks a lot!
>
>  2. *My other big concern is with respect to Theorem 1 in which they show the probability of exploding gradients. As far as I understand T1, the argument is that the gradient of a JSD divergence w.r.t a deterministic policy can explode. However, I'm not srue this can explain the experimental results, as this JSD is not the optimization target - instead an RL loss is being used.*
>
>  **Answer.** Indeed, it is difficult to demonstrate Theorem 1 experimentally, since it is hard to depict the exact optimal discriminator in experiments. Thus, we extend the scope of Theorem 1 to get Theorem 2, shown in pg 7. Actually, when the discriminator is set to be ranged in $(0,1)$,  DE-GAIL will also suffer from exploding gradients with some probability. Further, when the policy loss suffers from exploding gradients during many runs (*complete failure*), the discriminator in DE-GAIL degenerates to 0 or 1. This claim is more relevant to the invalidity in experiments and proposed in Theorem 2 in our revised version.
>
> Besides, we validate Theorem 2 through showing gradient performances in Fig. 6 and discriminator behaviors in Fig. 5. Please refer to Answer in the next question for the consistency between our theorems and experimental results.
>
>  3. *Lastly, more experimental results would be needed.*
>
>  **Answer.** Firstly, besides TD3-GAIL and SD3-GAIL, we additionally supply DDPG-GAIL in the revised version to strengthen our conclusion of theorem.
>
> Secondly, besides return curves, we complement gradient performances in Fig. 6 and discriminator behaviors in Fig. 5. Specifically, seeds 5, 7, 10 exhibit exploding gradient performances (left figure in Fig. 6) and degenerating discriminator behaviors (first row in Fig. 5), which are consistent with the invalid cases ($r\rightarrow 0$) in Fig. 4(a). With the aid of this complement, the exploding gradients in DE-GAIL is visualized and the logic relationship between our theory and the invalidity phenomenon in experiments is strengthened.

---

### Official Review · Reviewer_F1Ws · 2022-10-17

**Confidence:** 4
**Correctness:** 2
**Technical Novelty And Significance:** 2
**Empirical Novelty And Significance:** 2
**Recommendation:** 3

**Clarity, Quality, Novelty And Reproducibility:**

As discussed, the paper is difficult to follow in the current form, given the relatively poor writing quality.

**Strength And Weaknesses:**

Strength
1. The paper raises an interesting observation that deterministic policies may cause exploding gradient with AIL. If properly proven/demonstrated, the result would have significant impact on future method design.

Weakness
1. The paper is poorly structured and unclear. Many aspects of the proof are not sufficiently motivated or connected to the overall conclusion. For instance, it is unclear to me how Prop. 1 is related to Theorem 1. In addition, the notion of "non-confident" reward for ARIL has very little explanation (e.g. how it is related to Prop. 2).

2. The primary proof for the main claims is confusing and skip too many steps. Crucially, it is unclear how the exploding gradient is tied to $Pr(||\Sigma^{-1}(a_t - h(s_t))||_2 \ge C)$. In addition, it appears that $||\Sigma^{-1}(a_t - h(s_t))||_2 > 0$ since $L_2$ norm is positive with non-zero $\Sigma$. Therefore it is confusing why $Pr(||\Sigma^{-1}(a_t - h(s_t))||_2 \ge C)=0$ (Remark 2) for stochastic policies.

In addition, the proof relies on the assumption that for every gradient step, the discriminator is optimal. This is an unrealistic assumption and undermines the strength of the claim.

3. The empirical analysis only shows that deterministic policies fail for certain seed. If exploding gradient is indeed an issue as the theoretical results suggested, the authors should explicitly detect exploding gradients rather than just showing learning failures. On the other hand, a large number of runs must be repeated for stochastic polices (much more than 9 seeds, and many runs for each seed) to corroborate that stochastic policies have no exploding gradients, since it's impossible to empirically prove the negative (i.e. stochastic policies have no exploding gradient).

**Summary Of The Paper:**

The paper appears to show that deterministic policies may cause exploding gradient when used with adversarial imitation learning (AIL), despite their improved sample efficiency. At the same time, the paper appears to show that stochastic policies do not suffer from exploding gradients. This led to the conclusion that deterministic policies are "incompatible with AIL".

**Summary Of The Review:**

The paper is difficult to follow . While the observation/claim appears interesting, both the theoretical and empirical justification falls short. The manuscript may require significant revision before acceptance.

---

> ### Author Response · Authors · 2022-11-19
> **Answer to Reviewer F1Ws_1**
>
> 1. *The paper is poorly structured and unclear. Many aspects of the proof are not sufficiently motivated or connected to the overall conclusion. For instance, it is unclear to me how Prop. 1 is related to Theorem 1. In addition, the notion of "non-confident" reward for ARIL has very little explanation (e.g. how it is related to Prop. 2).*
>
>  **Answer.** We reconstruct the logic of the paper, and exhibit the architecture for our theoretical analysis (Fig. 2) for easy understanding. Major modifications are that the scope of Theorem 1 is extended to get Theorem 2, which can facilitate to demonstrate the consistency between experiments and theoretical analyses. The main modifications lie in:
>
> Firstly, Proposition 1 (original version) in the toy demo validates the existence of $r\rightarrow 0$ that emerged in experiments, and the toy demo is merely a descriptive example for the purpose of inducing our main result more naturally. The logic relationship between $r\rightarrow 0$ and exploding gradients theorem (*completely failure*) is listed below.
>
> Secondly, when the discriminator is set to be ranged in (0,1), DE-GAIL will also suffer from exploding gradients with some probability. This claim is more relevant to the invalidity in experiments and Theorem 2 (for $\tilde{D}\in (0,1)$) supplied additionally in our revised version.
>
> Thirdly, "non-confident" in the original version is a interpretive description in the sense of low sample efficiency, rather than strictly definition. In this revision, we omit this misunderstanding statement.
>
>  2. *The primary proof for the main claims is confusing and skip too many steps. Crucially, it is unclear how the exploding gradient is tied to $Pr(\|\Sigma^{-1}(a_{t}-h(s_{t}))\|_2\geq C)=0$. In addition, it appears that $\|\Sigma^{-1}(a_{t}-h(s_{t}))\|_2>0 $ since $L_2$ norm is positive with non-zero . Therefore it is confusing why $Pr(\|\Sigma^{-1}(a_{t}-h(s_{t}))\|_2\geq C)=0$ (Remark 2) for stochastic policies.*
>
>  **Answer.** Detailed proof is given in Appendix A.4 and Appendix A.5. Besides, for deterministic policy, i.e., $\Sigma \rightarrow 0$, we have $\Pr(\|\Sigma^{-1}(a_{t}-h(s_{t}))\|_2\geq C~\text{for any } C>0)\geq \Pr(\|a_{t}-h(s_{t})\|_2\geq C\|\Sigma\|_2~\text{for any } C>0)=\Pr(\Xi)$, and $\Pr(\Xi)\neq 0$. In contrast, for a Gaussian stochastic policy (fixed $\Sigma$), we have that $\|\hat{\nabla}_{h}D_{\rm JS}(\rho^{\pi_{h}},\rho^{\pi_{\rm E}})\|_2$ is bounded. Thus, when the discriminator is set to be optimal, the Gaussian stochastic policy in GAIL will not suffer from exploding gradients. Analogous conclusions can be drawn for non-Gaussian stochastic policies.
>
> *In addition, the proof relies on the assumption that for every gradient step, the discriminator is optimal. This is an unrealistic assumption and undermines the strength of the claim.*
>
>  **Answer.** In fact, not all gradient steps are trained under the optimal discriminator. Theorem 1 means the policy loss will suffer from exploding gradients once the discriminator achieves its optimal. To avoid such misunderstanding, we replace "When the discriminator is set to be optimal" with "When the discriminator achieves its optimal".
>
> In addition, we extend the theorem that when the discriminator is set to range in $(0, 1)$, DE-GAIL will also suffer from exploding gradients with some probability. Further, when the policy loss suffers from exploding gradients during many runs (*complete failure*), the discriminator in DE-GAIL degenerates to 0 or 1. This claim is more relevant to the invalidity in experiments and proposed in Theorem 2 in our revised version.
>
> Moreover, we also detect the exploding gradient performances in Fig. 6 and degenerating discriminator behaviors in Fig. 5 to validate our results.

---

> > ### Author Response · Authors · 2022-11-19
> > **Answer to Reviewer F1Ws_2**
> >
> >  3. *The empirical analysis only shows that deterministic policies fail for certain seed. If exploding gradient is indeed an issue as the theoretical results suggested, the authors should explicitly detect exploding gradients rather than just showing learning failures. On the other hand, a large number of runs must be repeated for stochastic polices (much more than 9 seeds, and many runs for each seed) to corroborate that stochastic policies have no exploding gradients, since it's impossible to empirically prove the negative (i.e. stochastic policies have no exploding gradient).*
> >
> >  **Answer.** Firstly, besides return curves, we complement gradient performances in Fig. 6 and discriminator behaviors in Fig. 5. With the aid of this complement, we visualize the exploding gradients in DE-GAIL and strengthen the logical relationship between our theory and the invalidity phenomenon in experiments.
> >
> > Secondly, we add several runs for both stochastic policies and deterministic policies in Fig. 4. Besides TD3-GAIL and SD3-GAIL, we additionally supply DDPG-GAIL (still exhibits instability) in the revised version to strengthen the accuracy of our conclusion.
> >
> > Thirdly, we supply more runs up to 11 seeds for stochastic policies; ST-GAIL does not occur exploding gradients (second row of FIg. 5 and right figure of Fig. 6) while seeds 5,7,10 of SD3-GAIL suffer from exploding gradients (first row of FIg. 5 and left figure of Fig. 6). Repeated experiments have been done by [1] (Fig. 2).
> >
> >  [1] Yirui Zhou, Yangchun Zhang, Xiaowei Liu, Wanying Wang, Zhengping Che, Zhiyuan Xu, Jian Tang, and Yaxin Peng. Generalization and computation for policy classes of generative adversarial imitation learning. In International Conference on Parallel Problem Solving from Nature, pp. 385–399, 2022.

---

> > ### Comment · Reviewer_F1Ws · 2022-11-25
> > **Thanks for the clarification**
> >
> > I thank the authors for the clarification and revised paper.
> >
> > From the revised paper, the key observation appears to be that the inverse of the covariance matrix $\Sigma^{-1}$ would explode during gradient computation, when $\Sigma$ tends to zero, which represents a deterministic policy.
> >
> > There are multiple issues with the above observation:
> > 1. Deterministic policies are generally not parametrized as a Gaussian distribution with $\Sigma \rightarrow 0$. When a policy is deterministic, $\Sigma$ is often not modeled and exploding gradients can't be attributed to it. In fact, the theoretical analysis may suggest the opposite, a stochastic policy with an learnable $\Sigma$ and tending to zero can lead to exploding gradient.
> > 2. Deterministic policy also requires exploration that adds noise to the output action. The policy plus exploration will thus still appear stochastic for GAIL optimization.
> > 3. The gradient computation from the theoretical analysis is unrealistic. The discriminator in practice can't be set to optimal (Theorem 1), or achieve the form in Theorem 2, since policy state-action distributions $\pi_E$ and $\pi_h$ are not directly accessible.
> > 4. Actual RL algorithms for improving policy may differ from a direct differentiation of the canonical loss function for GAIL.
> > 5. Many practical RL algorithms employ gradient clipping/scaling to prevent exploding gradient. It is unclear that how such practical techniques affect training stability.

---

> > > ### Author Response · Authors · 2022-12-07
> > > **Further Author Response to Reviewer F1Ws_1**
> > >
> > > [1] *Deterministic policies are generally not parameterized as a Gaussian distribution with $\Sigma \rightarrow 0$. When a policy is deterministic, $\Sigma$ is often not modeled and exploding gradients can't be attributed to it. In fact, the theoretical analysis may suggest the opposite, a stochastic policy with an learnable $\Sigma$ and tending to zero can lead to exploding gradient.*
> > >
> > > **Answer.** The noise (covariance matrix $\Sigma$) is indeed not learnable in the network of deterministic policy, refer to the training process of DDPG [1], TD3 [2] and SD3 [3]. In contrast, $\Sigma$ is learnable in stochastic policy, see [4,5]. Inspired by [6,7], arbitrary deterministic policy $h$ can be approximated by a family of Gaussian policies with mean $h$ and covariance $\Sigma$. Thus in this paper, we utilize Gaussian stochastic policy and simply decay $\Sigma$ towards zero to learn the deterministic policy $h$, thereby theoretically analysing the instability in DE-GAIL.
> > >
> > > [2] *Deterministic policy also requires exploration that adds noise to the output action. The policy plus exploration will thus still appear stochastic for GAIL optimization.*
> > >
> > > **Answer.** Thanks for your comment. Indeed, deterministic policy requires exploration that adds noise ($\Sigma'$) to the output action. Meanwhile, the covariance of Gaussian stochastic policy $\Sigma \rightarrow 0$ during learning the deterministic policy $h$ as aforementioned in the last question. In fact, the stochastic factors you mentioned have been concerned by using $\Xi_{1} $={$(s_{t},h(s_{t})):||h(s_{t})+\mathcal{N}(0,\Sigma')-a_{t}||\geq C ||\Sigma||$ for any $C>0$} characterized mismatching in our practice. During $\Sigma \rightarrow 0$, Theorem 1 and Theorem 2 both hold for $\Xi$ and $\Xi_{1}$ since the proofs of Theorem 1 and Theorem 2 only depends on $\Sigma \rightarrow 0$, regardless of introducing $\Sigma'$.
> > >
> > > [3] *The gradient computation from the theoretical analysis is unrealistic. The discriminator in practice can't be set to optimal (Theorem 1), or achieve the form in Theorem 2, since policy state-action distributions $\pi_{\rm E}$ and $\pi_{h}$ are not directly accessible.*
> > >
> > > **Answer.** In general, the discriminator has no explicit expression. In Eq. (7) of our revised version, when $\epsilon>0$ ($\epsilon<0$), the proportion of the expert state-action distribution $\rho^{\pi_{E}}$ is improved (decayed), and the discriminator value is higher (lower) than the optimal $\frac{\rho^{\pi_{E}}(s_{t},a_{t})}{\rho^{\pi_{E}}(s_{t},a_{t})+\rho^{\pi}(s_{t},a_{t})}$. In this way, Theorem 2 states an extension of Theorem 1 to a general case: the discriminator fluctuates near its optimal value. As noticed, a more rational form of the regular discriminator in Eq. (7) may be defined as $\tilde{D}(s_{t},a_{t})=\frac{(1+\epsilon_{1})\rho^{\pi_{E}}(s_{t},a_{t})}{(1+\epsilon_{1})\rho^{\pi_{E}}(s_{t},a_{t})+(1-\epsilon_{2})\rho^{\pi}(s_{t},a_{t})}$, where $\epsilon_{1}>-1, \epsilon_{2}<1$ or $\epsilon_{1}<-1, \epsilon_{2}>1$. Actually, the gradient estimator of the policy loss with respect to the policy’s parameter is at risk of exploding gradients as $\Sigma \rightarrow 0$ when the discriminator achieves its regular value.
> > >
> > > [1] Timothy P Lillicrap, Jonathan J Hunt, Alexander Pritzel, Nicolas Heess, Tom Erez, Yuval Tassa, David Silver, and Daan Wierstra. Continuous control with deep reinforcement learning. arXiv preprint arXiv:1509.02971, 2015.
> > >
> > > [2] Scott Fujimoto, Herke Hoof, and David Meger. Addressing function approximation error in actor-critic methods. In International Conference on Machine Learning, pp. 1587–1596, 2018.
> > >
> > > [3] Ling Pan, Qingpeng Cai, and Longbo Huang. Softmax deep double deterministic policy gradients. In Advances in Neural Information Processing Systems, volume 33, pp. 11767–11777, 2020.
> > >
> > > [4] Tuomas Haarnoja, Aurick Zhou, Pieter Abbeel, and Sergey Levine. Soft actor-critic: Off-policy maximum entropy deep reinforcement learning with a stochastic actor. In International Conference on Machine Learning, pp. 1861–1870, 2018a.
> > >
> > > [5] Tuomas Haarnoja, Aurick Zhou, Kristian Hartikainen, George Tucker, Sehoon Ha, Jie Tan, Vikash Kumar, Henry Zhu, Abhishek Gupta, Pieter Abbeel, et al. Soft actor-critic algorithms and applications. arXiv preprint arXiv:1812.05905, 2018b.
> > >
> > > [6] Santiago Paternain, Juan Andr´es Bazerque, Austin Small, and Alejandro Ribeiro. Stochastic policy gradient ascent in reproducing kernel hilbert spaces. IEEE Transactions on Automatic Control,66(8):3429–3444, 2020.
> > >
> > > [7] Guy Lever and Ronnie Stafford. Modelling policies in MDPs in reproducing kernel hilbert space. In Artificial Intelligence and Statistics, pp. 590–598, 2015.
> > >
> > > [8] Ilya Kostrikov, Kumar Krishna Agrawal, Debidatta Dwibedi, Sergey Levine, and Jonathan Tompson. Discriminator-actor-critic: Addressing sample inefficiency and reward bias in adversarial imitation learning. In International Conference on Learning Representations, 2019.

---

> > > > ### Author Response · Authors · 2022-12-07
> > > > **Further Author Response to Reviewer F1Ws_2**
> > > >
> > > > [4] *Actual RL algorithms for improving policy may differ from a direct differentiation of the canonical loss function for GAIL.*
> > > >
> > > > **Answer.** Our opinion on this issue is: Essentially, RL algorithms for improving policy and the gradient of the policy loss in GAIL are the same when the reward function is set as PLR $r=-\log(1-D)$. The reason is: Actually, the gradient estimator of the policy loss with respect to $h$ is unrelated to the first term. Under PLR, it is equal to policy gradients in RL algorithms. Thus, we consider PLR in our analysis.
> > > >
> > > > In addition, we incorporate the deterministic RL algorithms, such as DDPG, TD3 and SD3, into GAIL without modifying on the canonical loss function for GAIL. This paper investigates the mechanism on canonical DE-GAIL, which sheds light on extending to DE-GAIL with revisions, e.g., regularization term [8].
> > > >
> > > > [5] *Many practical RL algorithms employ gradient clipping/scaling to prevent exploding gradient. It is unclear that how such practical techniques affect training stability.*
> > > >
> > > > **Answer.** Thanks for your comment. [8] added gradient clipping to the actor network, which actually exhibited good performance (see Appendix B, Figure 8 in their work). Therefore, gradient clipping is an effective technique to improve training stability.
> > > >
> > > > Besides, in this paper, we clip the reward function, rather than gradient. Clipping reward shows its superiority in the stability of DE-GAIL but at the expense of lower sample efficiency. The experimental results in Appendix A.8 validate our analysis.
> > > >
> > > > [1] Timothy P Lillicrap, Jonathan J Hunt, Alexander Pritzel, Nicolas Heess, Tom Erez, Yuval Tassa, David Silver, and Daan Wierstra. Continuous control with deep reinforcement learning. arXiv preprint arXiv:1509.02971, 2015.
> > > >
> > > > [2] Scott Fujimoto, Herke Hoof, and David Meger. Addressing function approximation error in actor-critic methods. In International Conference on Machine Learning, pp. 1587–1596, 2018.
> > > >
> > > > [3] Ling Pan, Qingpeng Cai, and Longbo Huang. Softmax deep double deterministic policy gradients. In Advances in Neural Information Processing Systems, volume 33, pp. 11767–11777, 2020.
> > > >
> > > > [4] Tuomas Haarnoja, Aurick Zhou, Pieter Abbeel, and Sergey Levine. Soft actor-critic: Off-policy maximum entropy deep reinforcement learning with a stochastic actor. In International Conference on Machine Learning, pp. 1861–1870, 2018a.
> > > >
> > > > [5] Tuomas Haarnoja, Aurick Zhou, Kristian Hartikainen, George Tucker, Sehoon Ha, Jie Tan, Vikash Kumar, Henry Zhu, Abhishek Gupta, Pieter Abbeel, et al. Soft actor-critic algorithms and applications. arXiv preprint arXiv:1812.05905, 2018b.
> > > >
> > > > [6] Santiago Paternain, Juan Andr´es Bazerque, Austin Small, and Alejandro Ribeiro. Stochastic policy gradient ascent in reproducing kernel hilbert spaces. IEEE Transactions on Automatic Control,66(8):3429–3444, 2020.
> > > >
> > > > [7] Guy Lever and Ronnie Stafford. Modelling policies in MDPs in reproducing kernel hilbert space. In Artificial Intelligence and Statistics, pp. 590–598, 2015.
> > > >
> > > > [8] Ilya Kostrikov, Kumar Krishna Agrawal, Debidatta Dwibedi, Sergey Levine, and Jonathan Tompson. Discriminator-actor-critic: Addressing sample inefficiency and reward bias in adversarial imitation learning. In International Conference on Learning Representations, 2019.

---

### Official Review · Reviewer_59jo · 2022-10-26

**Confidence:** 4
**Correctness:** 3
**Technical Novelty And Significance:** 3
**Empirical Novelty And Significance:** 3
**Recommendation:** 6

**Clarity, Quality, Novelty And Reproducibility:**

* The writing is very obscure and difficult to understand.

*  It is interesting this paper explores the mechanism behind the generative adversarial imitation learning, which provide an important conclusion that instability is caused by deterministic policies, rather than GANs.

**Strength And Weaknesses:**

Pros:
* I believe that there is well backed motivation for work based off of the plentiful literature review.
*This article provides a theoretical basis for strengthening the study of learning stability and provides a plan for the design of future GAIL work.

Cons:
*This paper is only analyzed under the toy data set but does not test it under the actual scene data set.
* The composition of this article does not demonstrate carefully, and the writing is very obscure and difficult to understand.


**Summary Of The Paper:**

* This paper explores the mechanism behind the generative adversarial imitation learning, which provide an important conclusion that instability is caused by deterministic policies, rather than GANs.

* It provides some existing methods relieve exploding gradients, but at the expense of “non-confidence”, and ST-GAIL has advantages in mitigating instability.


**Summary Of The Review:**

I would tend to accept this paper as it is novel enough and supported by theoretical analysis.

---

> ### Author Response · Authors · 2022-11-19
> **Answer to Reviewer 59jo**
>
> 1. *This paper is only analyzed under the toy data set but does not test it under the actual scene data set.*
>
>  **Answer.** This paper theoretically analyses the instability in DE-GAIL. The theorems are validated by the exploding gradient performances in Fig. 6 and degenerating discriminator behaviors in Fig. 5 in MuJoCo experimentally. MuJoCo is a representative environment that is widely used in reinforcement learning. In future work, we will explore an algorithm that improves the performance of DE-GAIL based on our theorems, and test it under other benchmarks and actual scene data sets.
>
>  2. *The composition of this article does not demonstrate carefully, and the writing is very obscure and difficult to understand.*
>
>  **Answer.**  We appreciate your suggestion.  We have checked the grammar using the software Grammarly and rewritten the unclear statements.  In addition, we reconstruct the logic of the paper and exhibit the architecture for our theoretical analysis (Fig. 2) for easy understanding. Major modifications are that the scope of Theorem 1 is extended to get Theorem 2, which facilitates the demonstration of the consistency between experiments and theoretical analyses.

---

### Official Review · Reviewer_EU3W · 2022-10-27

**Confidence:** 3
**Correctness:** 1
**Technical Novelty And Significance:** 2
**Empirical Novelty And Significance:** 1
**Recommendation:** 3

**Clarity, Quality, Novelty And Reproducibility:**

While the arguments made in the paper are quite original, I believe that the quality and clarity of the paper are not good enough.

**Strength And Weaknesses:**

Strength:
- this paper addresses a less explored problem of theoretically analyzing the "instability" of previous GAIL variants.

Weaknesses:
- bad writing quality:

There are a large number of grammatical errors and word misusages. The paper also refers to a number of other algorithms without any detailed explanation - e.g. TD3-GAIL, SD3-GAIL, and combination reward function. How they are defined is quite important in this analysis paper, so I think it had to be described in the paper for completeness. There are many theoretical results, but their implications are not well explained. These issues are combined to make the paper extremely hard to read.

- not convincing analyses:

1. The paper starts by arguing that some algorithms show pathological behavior (of high instability) when PLR and deterministic policy is combined. However, the paper simply makes the comparisons between SD3-GAIL, TD3-GAIL, and TSSG; by comparing different algorithms, there are so many different factors other than PLR and deterministic policy that affects the performance, and it is not accurate to make the hypothesis out of these comparisons. I believe the authors should have compared the consequence of different reward functions and policy choices in the same algorithm. Furthermore, it would have been better if other metrics other than simple returns are compared, e.g. gradient variance, minimum learned reward, etc, to show the validity of further technical analyses.

2. In proposition 1, the authors argue that PLR is bad because, in this toy domain, the reward can approach 0 as the number of states grows to infinity. Why is that a problem? It will be fine unless we have $r=0$ for all state actions, and for a certain number of states $|\mathcal{S}|$ we will always have positive rewards for some state actions. And it seems to be natural to have a small reward if we have such long expert trajectories.

3. In theorem 1, the authors argue that the variance of gradient explodes as a variance of a policy approaches 0. However, it is not a new analysis and it is widely known if we have a low-variance stochastic policy, the gradient variance of usual policy gradient algorithms will explode, e.g. in REINFORCE algorithm. And it is also well known that DPG does not suffer from such a problem since it has defined the policy as deterministic in the first place. Therefore, for me, the author's argument on theorem 1 of saying GAIL with deterministic policies is unstable is not convincing unless any empirical evidence is provided, e.g. the actual variance of gradient during training.

- the result is not very significant:

The paper does not propose any new algorithm based on the analysis. Since most of the findings in this paper do not seem to be very significant to me, I cannot highly value the contribution of the paper.

**Summary Of The Paper:**

This paper aims to show that using deterministic policy in a generative adversarial imitation learning algorithm is not proper, and results in a significant instability. This paper does the following:
- empirical comparisons between SD3-GAIL, TD3-GAIL, TSSG
- show a toy domain and prove that PLR approaches 0 as the size of state space increases
- variance of the gradient of GAIL objective explodes in mismatched cases if we use deterministic policy
- if we use CR instead of PLR, such a problem is addressed with some cost.


**Summary Of The Review:**

Due to the weaknesses I listed above: low writing quality, not convincing analysis, and lack of algorithm as a result of the analysis, I recommend rejection of the paper.

---

> ### Author Response · Authors · 2022-11-19
> **Answer to Reviewer EU3W 1**
>
> 1. *bad writing quality:
> There are a large number of grammatical errors and word misusages. The paper also refers to a number of other algorithms without any detailed explanation - e.g. TD3-GAIL, SD3-GAIL, and combination reward function. How they are defined is quite important in this analysis paper, so I think it had to be described in the paper for completeness. There are many theoretical results, but their implications are not well explained. These issues are combined to make the paper extremely hard to read.*
>
>  **Answer.** We have done our best to improve our English in the revised version.
>
> GAIL introduces the concepts of discriminator $D$  to RL algorithms. The original GAIL is established by the trust region policy optimization (TRPO) algorithm. To suit the minimax objective optimization problem, the reward function is set as (PLR) $r=-\log(1-D)$. The SD3-GAIL, TD3-GAIL and DDPG-GAIL in our work are set up by replacing TRPO to SD3, TD3 and DDPG with unchanged $r=-\log(1-D)$. TD3-GAIL and DDPG-GAIL have been investigated with some modifications on the reward function or its derivatives [1,2]. These modificiations improve the bad behaviors (training failure) for DE-GAIL when adapting original PLR, which inhibits the investigation for the underlying mechanism of such failure. That is to say,  what will happen to these DE-GAIL algorithms if the reward function is exactly $r=-\log(1-D)$?
>
> We have added detailed implications with respect to our theorems in Section 4.2. Besides, we have also supplied the architecture of our theoretical analysis (Fig. 2) for easy understanding, explaining the connection among each conclusion.  Actually, we find that when the discriminator is set to range in $(0, 1)$, DE-GAIL will also suffer from exploding gradients with some probability. This claim is more relevant to the invalidity in experiments ($r\rightarrow 0$ in (a), (b), (d) in Fig. 4) and proposed in Theorem 2 in our revised version. Further, when the policy loss suffers from exploding gradients during many runs  (*complete failure*), the discriminator in DE-GAIL degenerates to 0 or 1.
>
>  2. *not convincing analyses:*
>  2.1. *The paper starts by arguing that some algorithms show pathological behavior (of high instability) when PLR and deterministic policy is combined. However, the paper simply makes the comparisons between SD3-GAIL, TD3-GAIL, and TSSG; by comparing different algorithms, there are so many different factors other than PLR and deterministic policy that affects the performance, and it is not accurate to make the hypothesis out of these comparisons. I believe the authors should have compared the consequence of different reward functions and policy choices in the same algorithm. Furthermore, it would have been better if other metrics other than simple returns are compared, e.g. gradient variance, minimum learned reward, etc, to show the validity of further technical analyses.*
>
>  **Answer.** Thanks for your comments.
> Factors other than PLR affect the investigation for the underlying mechanism of training failure in DE-GAIL. In addition, the original minimax optimization problem in GAIL may be changed by these factors, thereby the structure of GAIL is shifted. In this paper, we focus on the original GAIL where the reward is set as PLR $r(s,a)=-\log(1-D(s,a))$. For the comparison among different reward functions, the claim that CR-DE-GAIL exactly performs better than PLR-DE-GAIL has been validated empirically in Fig. 5 of [1]. In this paper, we verify it in theory. The theoretical insight for training failure in DE-GAIL under exact PLR is provided in Section 4.2.
>
> In order to enhance the accuracy of our conclusions, additionally deep deterministic policy gradient (DDPG) is introduced into GAIL. DDPG-GAIL exhibits the same instability as the experiment results in SD3-GAIL and TD3-GAIL.
>
> Besides the gradient performance in Fig. 6, we supply discriminator behaviors in Fig. 5 additionally to validate our result.
>
> [1] Ilya Kostrikov, Kumar Krishna Agrawal, Debidatta Dwibedi, Sergey Levine, and Jonathan Tompson. Discriminator-actor-critic: Addressing sample inefficiency and reward bias in adversarial imitation learning. In International Conference on Learning Representations, 2019.
>
> [2] Guoyu Zuo, Kexin Chen, Jiahao Lu, and Xiangsheng Huang. Deterministic generative adversarial imitation learning. Neurocomputing, 388:60–69, 2020.
>
> [3] Timothy P Lillicrap, Jonathan J Hunt, Alexander Pritzel, Nicolas Heess, Tom Erez, Yuval Tassa, David Silver, and Daan Wierstra. Continuous control with deep reinforcement learning. arXiv preprint arXiv:1509.02971, 2015.

---

> > ### Author Response · Authors · 2022-11-19
> > **Answer to Reviewer EU3W 2**
> >
> > 2.2. *In proposition 1, the authors argue that PLR is bad because, in this toy domain, the reward can approach 0 as the number of states grows to infinity. Why is that a problem? It will be fine unless we have $r=0$ for all state actions, and for a certain number of states $\mathcal{S}$ we will always have positive rewards for some state actions. And it seems to be natural to have a small reward if we have such long expert trajectories.*
> >
> >  **Answer.** The toy demo is served as an illustrative example to interpret the existence of invalid cases ($r\rightarrow 0$), which naturally induces our main result on exploding gradients in Section 4.2. In fact, for the state $s_{t}$ ($0\leq t< 3g/4$), the action $a_{t\rightarrow t+1}$ occurs 3 times while others occur at most twice in the expert demonstration. For the reason that $D^{\ast}(s,a)=\frac{\rho^{\pi_{\rm E}}(s,a)}{\rho^{\pi_{\rm E}}(s,a)+\rho^{\pi}(s,a)}=\frac{1}{1+\frac{\rho^{\pi}(s,a)}{\rho^{\pi_{\rm E}}(s,a)}}$, $D^{\ast}(s,a)$ is monotonically increasing with respect to $\rho^{\pi_{\rm E}}(s,a)$. Thus, the more frequent expert state-action pair tends to attain the higher value of the discriminator, thereby leading to higher PLR. Meanwhile, policy training in RL aims to maximize the expected reward-to-go. Therefore, the most frequent action $a_{t\rightarrow t+1}$ is the best choice for a deterministic learned policy under each $s_{t}$. Note that under the deterministic policy, PLR of the best action approaches 0. As a result, $r\rightarrow 0$ for all state-action pairs, thereby resulting in invalidity in the evaluation of the learned policy. Besides, we put this part into Appendix A.2 due to the limited space in the main part.
> >
> >  2.3. *In theorem 1, the authors argue that the variance of gradient explodes as a variance of a policy approaches 0. However, it is not a new analysis and it is widely known if we have a low-variance stochastic policy, the gradient variance of usual policy gradient algorithms will explode, e.g. in REINFORCE algorithm. And it is also well known that DPG does not suffer from such a problem since it has defined the policy as deterministic in the first place. Therefore, for me, the author's argument on theorem 1 of saying GAIL with deterministic policies is unstable is not convincing unless any empirical evidence is provided, e.g. the actual variance of gradient during training.*
> >
> >  **Answer.** Incorporating the model-free RL algorithm into the framework of GAIL could induce new problems that do not occur in the RL algorithm itself, one can refer to [1].
> >
> > According to the reviewers’ suggestions, we have added the experiment of DDPG-GAIL, which suffers from the same unstable training process (with PLR) as in SD3-GAIL and TD3-GAIL, see Fig. 3 and Fig. 4(d). Note that DDPG is a deterministic policy algorithm with a stable learning process in model-free RL [3]. The two corner behaviors verify our theorems.
> >
> > Besides return curves for each method, we also detect the exploding gradient performances in Fig. 6 and the degenerating behaviors of the discriminator in Fig. 5, which also validate our results.
> >
> >  [1] Ilya Kostrikov, Kumar Krishna Agrawal, Debidatta Dwibedi, Sergey Levine, and Jonathan Tompson. Discriminator-actor-critic: Addressing sample inefficiency and reward bias in adversarial imitation learning. In International Conference on Learning Representations, 2019.
> >
> >  [2] Guoyu Zuo, Kexin Chen, Jiahao Lu, and Xiangsheng Huang. Deterministic generative adversarial imitation learning. Neurocomputing, 388:60–69, 2020.
> >
> >  [3] Timothy P Lillicrap, Jonathan J Hunt, Alexander Pritzel, Nicolas Heess, Tom Erez, Yuval Tassa, David Silver, and Daan Wierstra. Continuous control with deep reinforcement learning. arXiv preprint arXiv:1509.02971, 2015.

---

> > > ### Author Response · Authors · 2022-11-19
> > > **Answer to Reviewer EU3W 3**
> > >
> > > 3. *the result is not very significant:
> > > The paper does not propose any new algorithm based on the analysis. Since most of the findings in this paper do not seem to be very significant to me, I cannot highly value the contribution of the paper.*
> > >
> > >  **Answer.** Our theoretical analysis describes that a smaller discriminator interval of outliers (Definition 2) will decrease the probability of gradient explosion. Based on this result, we propose a new algorithm, i.e., clipping the reward that leads to the outlier of the discriminator. The preliminary results of our new algorithm have been added in the revised version. In our experiment, clipping reward shows its superiority in the stability of DE-GAIL. Detailed results in three MuJoCo environments can refer to Appendix A.8, Fig. 10.

---

### Official Review · Reviewer_Qudi · 2022-11-03

**Confidence:** 3
**Correctness:** 3
**Technical Novelty And Significance:** 4
**Empirical Novelty And Significance:** 3
**Recommendation:** 5

**Clarity, Quality, Novelty And Reproducibility:**

**Clarity**: poor

Below are the questions related to the clarity of the paper:

1. In Figure 6, adding the legend for the red line and also adding x-axis and y-axis labels will make the clarity of the paper higher. And rather than writing $a_{t1}$ and $a_{t2}$ in the figure, putting markers and adding legends showing that they are optimal and suboptimal actions would be more informative to the readers.

2. On pg 7, I found this sentence hard to understand: “The learned policy is exposure to be transferred to the mismatching case due to the limited area in the sub-optimal matching compared with the optimal”. I’d appreciate elaboration on the sentence.

3. I was unable to find information on how Fig 6 was drawn. Was it drawn as an illustrative example? or, is it an empirical result?

4. On pg 7, the authors mentioned that “The threshold of matching is set as 0.035.”. Could you elaborate on how 0.035 was computed? Also, the “threshold” was never mentioned before. What is the “threshold” and where is it applied?

5. In Appendix A.2 (proof of Theorem 1), the authors cited (Guan et al., 2021a) for the gradient of the JS divergence with respect to h, but I was not able to find the contents related to the gradients of JS divergence in the paper. Is the paper cited for policy gradient? If so, elaboration on the derivation of $\hat{\nabla}_h   D_J   (\rho^{\pi_h}, \rho^{\pi_E} )$ ($D_J$: JS divergence) would help the readers to understand.

6. Is the probabilistic lower bound related to the lower bound in Remark 2? If not, where is it mentioned in the paper?

**Reproducibility** : good
The details of the experiments for Fig 2 and 3 are mentioned in the paper

**Quality** : poor
Due to the lack of details in Fig 6 and hard-to-understand sentences.

**Novelty** : good




**Strength And Weaknesses:**

**Strength**
1. The authors theoretically show that the deterministic learned policy brings instability in training.

2. The authors theoretically show that the problem can be alleviated by using the modified reward function of AIRL.

**Weaknesses and Questions**

1. There are no experiments that validate or support Proposition 1. The experimental results in Fig 1~3 are made from environments with continuous state and action spaces while Proposition 1 is about environments with finite action space. If the authors could empirically validate Proposition 1, that would strengthen the paper.

2. There are no experiments that validate Proposition 2. The empirical results shown in the paper are only about PLR-DE-GAIL. The results of CR-DE-GAIL should be compared with that of PLR-DE-GAIL.

3. Need more strong empirical validation for Theorem 1. Fig 2 and 3 may show instability in training DE-GAIL, but they don’t empirically show that the exploding gradients problem actually happened and that the problem is caused by the instability. If the authors could show that the low-performing seeds in Figure 3 indeed suffered from the exploding gradients while the other seeds did not, that would validate Theorem 1 strongly.

4. The statement In pg 6: “For any state $s_t$, the action $a_{t \rightarrow t+1}$ occurs 3 times while others occur at most twice in the expert demonstration.” seems to be wrong. If you follow the pseudo-code in Figure 4, when the agent is in $s_{\frac{3g}{4}+1}, s_{\frac{3g}{4}+2}, \cdots, s_{g-1}$ the agent never executes the action $a_{t \rightarrow t+1}, \forall t\in\{ \frac{3g}{4}+1, \frac{3g}{4}+2, \cdots, g-1  \}$.

5. On pg 7, authors mentioned that “The learned policy is exposure to be transferred to the mismatching case due to the limited area in the sub-optimal matching compared with the optimal; see seed 1 in Fig. 3(a) for instance”. But I think you cannot guarantee that the result from seed 1 in Fig.3 (a) is such a case. Could you show that the seed 1 experiment shows results similar to Fig 6?

6. Proposition 1: The derivation of the “length of the expert trajectory” seems to be wrong. In Appendix A.1 in the proof of Proposition 1, the authors derived this equation :  $N=2\left(\frac{g}{4}+\left(\frac{g}{4}+1\right)+\cdots+\left(\frac{g}{4}+\frac{g}{4}-1\right)\right)+2\left(\frac{g}{2}+1\right) \frac{g}{2}-1+\frac{3 g}{4}=\frac{11 g^2+24 g-16}{16}$. I may be wrong, but the derivation is different from mine. For the first for loop in the pseudo-code of Figure 4, the sub-trajectory length is $2\left(\frac{g}{4}+\left(\frac{g}{4}+1\right)+\cdots+\left(\frac{g}{4}+\frac{g}{4}\right)\right)+\frac{3 g}{4}$ where the first term comes from executing actions $a_{i\rightarrow j}, a_{j\rightarrow i}$, and the last term comes from executing the action $a_{i\rightarrow i+1}$. For the second for loop, since the agent only traverses between $s_{\frac{3g}{4}}$ and $s_{\frac{g}{2}}, s_{\frac{g}{2}+1}, \cdots, s_{g-1}$ except $s_{3g/4}$, the length of sub-trajectory due to the second for loop should be $2\left( (g-1)-g/2 \right)$. For the last line, the sub-trajectory of length 1 comes from executing the action  $a_{3g/4 \rightarrow g}$. By summing them up, I got $(g^2 + 36g)/16$.

7. According to “Definition 1” on pg 7,   state-action pairs are in a mismatch when $|| \Sigma ||_2 \rightarrow 0$ and $a_t \neq h(s_t)$. And the pairs match when $a_t = h(s_t)$. But Fig 6 draws actions in the neighborhood of a_t1 to show the matched case. And the paragraph under Fig 6 mentions that the matched case is made by having actions in the neighborhood of a_t1.

8. In Appendix A.2 (proof of Theorem 1), the authors derived $\nabla_h \pi_h(a \mid s)=\pi_h(a \mid s) \kappa(s, \cdot) \boldsymbol{\Sigma}^{-1}(a-h(s))$. Why is there $\kappa$ when you define $\pi_h$ as in the equation at the bottom of pg 6?

9. In Remark 2, since $|| \Sigma ||^{-1}_2 ||a_t-h(s_t)||_2 \ge || \Sigma^{-1} (a_t-h(s_t))||_2$, shouldn't be the inequality between probabilities (probability mentioned in Theorem 1 and $\operatorname{Pr}(\Xi)$) the other way around?


**Summary Of The Paper:**

The paper is about the instability in training when a deterministic learned policy is used in generative adversarial imitation learning (DE-GAIL) and how to alleviate the problem. The authors first show that the training of DE-GAIL is empirically unstable in Mujoco environments that have continuous action and state spaces. Then, they theoretically show that the reward of DE-GAIL becomes zero when the deterministic policy is used in finite MDP and when the cardinality of the state space goes to infinity in Proposition 1. Also, they argue that the instability is mainly due to the reward becoming zero. For MDPs with continuous action spaces, they theoretically show that DE-GAIL suffers the problem of exploding gradients due to a mismatch between actions from the expert and the learned policy. Furthermore, they use Lemma 1 to show that having a mismatch between the actions from the expert and the learned policy is equivalent to having an optimal discriminator function value of one which is also equivalent to having an infinite reward. Lastly, in Proposition 2, they prove that using CR instead of PLR can alleviate the problem. This is done by showing that PLR has a value larger than or equal to that of CR when their discriminator values are the same.

**Summary Of The Review:**

The paper presents a novel theoretical analysis of the instability in the training of DE-GAIL and also theoretically shows how the modified reward function of AIRL can alleviate the problem. However, there seem to be some cases where statements or derivations are wrong, Furthermore, the paper lacks experiments that support propositions and theorems.

---

> ### Author Response · Authors · 2022-11-19
> **Answer to Reviewer Qudi 1**
>
> 1. *``There are no experiments that validate or support Proposition 1. The experimental results in Fig. 1$\sim$3 are made from environments with continuous state and action spaces while Proposition 1 is about environments with finite action space. If the authors could empirically validate Proposition 1, that would strengthen the paper. "*
>
> **Answer.** Thank you for your suggestion. Indeed, we have not put supportive experiments to Prop. 1. The reasons are as follows. The toy demo served as an illustrative example to interpret the existence of invalid cases ($r\rightarrow 0$), which intuitively inspires the Theorem of exploding gradients in Section 4.2. Actually, it is more common to utilize the performance in classical MuJoCo environments as the measure of the main results (see [1,2]). So in MuJoCo environments, we add many gradient performances in Fig. 6 and discriminator behaviors in Fig. 5.
>
> 2. *``There are no experiments that validate Proposition 2. The empirical results shown in the paper are only about PLR-DE-GAIL. The results of CR-DE-GAIL should be compared with that of PLR-DE-GAIL. "*
>
> **Answer.** The comparison of  CR-DE-GAIL and PLR-DE-GAIL has been done in Fig. 5 of [1]. The conclusion is that CR-DE-GAIL exactly performs better than PLR-DE-GAIL. In this paper, we proof it theoretically.
>
> 3. *``Need more strong empirical validation for Theorem 1. Fig. 2 and 3 may show instability in training DE-GAIL, but they don’t empirically show that the exploding gradients problem actually happened and that the problem is caused by the instability. If the authors could show that the low-performing seeds in Figure 3 indeed suffered from the exploding gradients while the other seeds did not, that would validate Theorem 1 strongly. "*
>
> **Answer.** Thank you for your suggestion. Besides the gradient performance, we have supplied discriminator behaviors additionally (see Fig. 5 in the revision).
>
> 4. *"The statement In pg 6:"For any state $s_{t}$, the action $a_{t\rightarrow t+1}$ occurs 3 times while others occur at most twice in the expert demonstration." seems to be wrong. If you follow the pseudo-code in Figure 4, when the agent is in $s_{\frac{3g}{4}+1}$, $s_{\frac{3g}{4}+2},...,s_{g-1}$ the agent never executes the action $a_{t\rightarrow t+1}$, $\forall t \in \frac{3g}{4}+1,\frac{3g}{4}+2,...,g-1$. "*
>
> **Answer.** Thank you for your careful check! We have revised it in the revision.
>
> 5. *``On pg 7, authors mentioned that “The learned policy is exposure to be transferred to the mismatching case due to the limited area in the sub-optimal matching compared with the optimal; see seed 1 in Fig. 3(a) for instance”. But I think you cannot guarantee that the result from seed 1 in Fig. 3(a) is such a case. Could you show that the seed 1 experiment shows results similar to Fig. 6? "*
>
> **Answer.** Thanks! We have rectified the misunderstanding point. Fig. 6 is only a descriptive example of mismatching and matching aiming to assist understanding for readers, rather than an empirical result as Fig. 3(a). We put this part into Appendix A.3 due to the limited space in the main part.
>
> 6. *``Proposition 1: The derivation of the "length of the expert trajectory" seems to be wrong. In Appendix A.1 in the proof of Proposition 1, the authors derived this equation: $N=2(\frac{g}{4}+(\frac{g}{4}+1)+...+(\frac{g}{4}+\frac{g}{4}-1))+2(\frac{g}{2}+1)\frac{g}{2}-1+\frac{3g}{4}=\frac{11g^{2}+24g-16}{16}$. I may be wrong, but the derivation is different from mine. For the first for loop in the pseudo-code of Figure 4, the sub-trajectory length is $2(\frac{g}{4}+(\frac{g}{4}+1)+...+(\frac{g}{4}+\frac{g}{4}))+\frac{3g}{4}$ where the first term comes from executing actions $a_{i\rightarrow j},a_{j\rightarrow i}$, and the last term comes from executing the action $a_{i\rightarrow i+1}$. For the second for loop, since the agent only traverses between $s_{3g/4}$ and $s_{g/2},s_{g/2+1},...,s_{g-1}$ except $s_{3g/4}$, the length of sub-trajectory due to the second for loop should be $2((g-1)-g/2)$. For the last line, the sub-trajectory of length 1 comes from executing the action $a_{3g/4\rightarrow g}$. By summing them up, I got $(g^{2}+36g)/16$. "*
>
> **Answer.** We have checked it carefully, and find the original version is correct. A detailed explanation is in Appendix A.2, pg 12.
>
> [1] Ilya Kostrikov, Kumar Krishna Agrawal, Debidatta Dwibedi, Sergey Levine, and Jonathan Tompson. Discriminator-actor-critic: Addressing sample inefficiency and reward bias in adversarial imitation learning. In International Conference on Learning Representations, 2019.
>
> [2] Tian Xu, Ziniu Li, and Yang Yu. Error bounds of imitating policies and environments. In Advances in Neural Information Processing Systems, volume 33, pp. 15737–15749, 2020.

---

> > ### Author Response · Authors · 2022-11-19
> > **Answer to Reviewer Qudi 2**
> >
> > 7. *According to Definition 1 on pg 7, state-action pairs are in a mismatch when $\|\Sigma\|_{2}\rightarrow 0$
> >
> > and $a_{t}\neq h(s_{t})$. And the pairs match when $a_{t}=h(s_{t})$. But Fig. 6 draws actions in the neighborhood of $a_{t1}$ to show the matched case. And the paragraph under Fig. 6 mentions that the matched case is made by having actions in the neighborhood of $a_{t1}$. *
> >
> >  **Answer.** The mismatched and matched state-action pairs are defined by the neighborhood, rather than whether $a_{t}=h(s_{t})$. Actually, when the mismatched and matched state-action pairs are considered under DE-GAIL ($\|\Sigma\|_{2}\rightarrow 0$), defined by neighborhood is more appropriate. Details can refer to Definition 1.
> >
> >  8. *``In Appendix A.2 (proof of Theorem 1), the authors derived $\triangledown_{h}\pi_{h}(a|s)=\pi_{h}(a|s)\kappa(s,\cdot)\Sigma^{-1}(a-h(s))$. Why is there $\kappa$ when you define $\pi_{h}$ as in the equation at the bottom of pg 6? "*
> >
> >  **Answer.** Thank you for your suggestion. The relationship between $h(\cdot)$ and $\kappa$ is supplied in pg 6. The function $h(\cdot)$ is an element of an RKHS $\mathcal{H}_{\kappa}$, $h(\cdot)=\sum_{i}\kappa(s_{i},\cdot)a_{i}\in \mathcal{H}_{\kappa}$, where $s_{i}\in \mathcal{S}$ and $a_{i}\in \mathcal{A}$.
> >
> >  9. In Remark 2, since $\|\Sigma\|_2^{-1} \|a_{t}-h(s_{t})\|_2\geq |\Sigma^{-1}(a_{t}-h(s_{t}))|_2$,
> >
> > shouldn't be the inequality between probabilities (probability mentioned in Theorem 1 and ${\rm Pr}(\Xi)$) the other way around?
> >
> >  **Answer.** Actually, the inequality is obtained by $\|a_{t}-h(s_{t})\|_2=\| \bm{\Sigma} \bm{\Sigma}^{-1} (a_{t}-h(s_{t}))\|_2 \leq \| \bm{\Sigma} \|_2\|\bm{\Sigma}^{-1} (a_{t}-h(s_{t}))\|_2$. In fact, $\|\bm{\Sigma}\|_{2}^{-1}\neq \|\Sigma^{-1}\|_{2}$.
> >
> >  10. *``In Figure 6, adding the legend for the red line and also adding x-axis and y-axis labels will make the clarity of the paper higher. And rather than writing at1 and at2 in the figure, putting markers and adding legends showing that they are optimal and suboptimal actions would be more informative to the readers. "*
> >
> >  **Answer.** Thank you for your suggestion. We have followed your suggestion by adding the legends to explain Figure 6 clearly.
> >
> >  11. *``On pg 7, I found this sentence hard to understand: "The learned policy is exposure to be transferred to the mismatching case due to the limited area in the sub-optimal matching compared with the optimal". I’d appreciate elaboration on the sentence. "*
> >
> >  *``I was unable to find information on how Fig. 6 was drawn. Was it drawn as an illustrative example? or, is it an empirical result? "*
> >
> >  *``On pg 7, the authors mentioned that "The threshold of matching is set as 0.035.". Could you elaborate on how 0.035 was computed? Also, the "threshold" was never mentioned before. What is the "threshold" and where is it applied? "*
> >
> > **Answer.** It is a descriptive example, rather than an empirical result. We rewrite this part in the revision. Without loss of generality, the threshold of matching is set as 0.035. The trajectories used to train the learned policies are shown in red curves. The matching is that $h(s_{t})$ lies in the neighborhood of $a_{t}$, which has a high probabilistic density in the expert demonstration.
> >
> >  12. *``In Appendix A.2 (proof of Theorem 1), the authors cited (Guan et al., 2021a) for the gradient of the JS divergence with respect to h, but I was not able to find the contents related to the gradients of JS divergence in the paper. Is the paper cited for policy gradient? If so, elaboration on the derivation of $\hat{\triangledown}_{h} D_J(\rho^{\pi_h},\rho^{\pi_E})$ ($D_J$: JS divergence) would help the readers to understand. "*
> >
> >  **Answer.** Thank you for your insightful suggestion. Firstly, the purpose of the citation is to interpret the independence between the state distribution $d^{\pi}(s)$ and the policy, which is a key condition in our proof. Secondly, we complement detailed proof in the revised version (Appendix A.4).
> >
> >  13. *``Is the probabilistic lower bound related to the lower bound in Remark 2? If not, where is it mentioned in the paper? "*
> >
> >  **Answer.** Yes, we complement the relationship in Remark 2 (Now Remark 1).

---

### Author Response · Authors · 2022-11-19
**Instability in Generative Adversarial Imitation Learning with Deterministic Policy**

The authors appreciate the reviewers and chairs for their very helpful comments.

We change the original title to "Instability in Generative Adversarial Imitation Learning with Deterministic Policy". All issues from the reviewer have been addressed in this revised paper.

Below are our point-to-point replies to reviewers' comments.

---

### Decision · Program_Chairs · 2023-01-20

**Decision:**

Reject

**Justification For Why Not Higher Score:**

overall low score - contribution is low and writing needs to be improved

**Justification For Why Not Lower Score:**

N/A

**Metareview: Summary, Strengths And Weaknesses:**

This paper presents the ill-posedness of using deterministic policies for the family of adversarial imitation learning algorithms. Albeit this is an important problem to be addressed and this work could be taking an important step towards a more robust algorithm, this paper has room for improvement. Reviewers all commented that the writing needs to be improved, and the idea has to be validated on more challenging tasks. I hope the comments below help the authors revise the paper for a future conference.